# Advances in Concentration Gradient Generation Approaches in a Microfluidic Device for Toxicity Analysis

**DOI:** 10.3390/cells11193101

**Published:** 2022-10-01

**Authors:** Nicole M. E. Valle, Mariana P. Nucci, Arielly H. Alves, Luiz D. Rodrigues, Javier B. Mamani, Fernando A. Oliveira, Caique S. Lopes, Alexandre T. Lopes, Marcelo N. P. Carreño, Lionel F. Gamarra

**Affiliations:** 1Hospital Israelita Albert Einstein, São Paulo 05652-000, Brazil; 2Pontifícia Universidade Católica de São Paulo, São Paulo 01303-050, Brazil; 3LIM44—Hospital das Clínicas da Faculdade Medicina da Universidade de São Paulo, São Paulo 05403-000, Brazil; 4Departamento de Engenharia de Sistema Eletrônicos, Escola Politécnica, Universidade de São Paulo, São Paulo 05508-010, Brazil

**Keywords:** microfluidic device, microdevice, concentration gradient generator, CGG, toxicity, drug screening, microdevice gradient generator

## Abstract

This systematic review aimed to analyze the development and functionality of microfluidic concentration gradient generators (CGGs) for toxicological evaluation of different biological organisms. We searched articles using the keywords: concentration gradient generator, toxicity, and microfluidic device. Only 33 of the 352 articles found were included and examined regarding the fabrication of the microdevices, the characteristics of the CGG, the biological model, and the desired results. The main fabrication method was soft lithography, using polydimethylsiloxane (PDMS) material (91%) and SU-8 as the mold (58.3%). New technologies were applied to minimize shear and bubble problems, reduce costs, and accelerate prototyping. The Christmas tree CGG design and its variations were the most reported in the studies, as well as the convective method of generation (61%). Biological models included bacteria and nematodes for antibiotic screening, microalgae for pollutant toxicity, tumor and normal cells for, primarily, chemotherapy screening, and Zebrafish embryos for drug and metal developmental toxicity. The toxic effects of each concentration generated were evaluated mostly with imaging and microscopy techniques. This study showed an advantage of CGGs over other techniques and their applicability for several biological models. Even with soft lithography, PDMS, and Christmas tree being more popular in their respective categories, current studies aim to apply new technologies and intricate architectures to improve testing effectiveness and reduce common microfluidics problems, allowing for high applicability of toxicity tests in different medical and environmental models.

## 1. Introduction

The toxicological assessment of chemicals, pharmaceuticals, food and food ingredients, cosmetics, and industrial products has significantly advanced due to scientific and technological developments. New techniques, such as the promising alternative of human-cell-seeded organ-on-chips for acute systemic toxicity, as well as in silico approaches, have been replacing conventional techniques, for example, tests which use LD_50_ as their main parameter, requiring a great number of animals to determine the chemical dose able to achieve 50 percent of deaths [1]. After the 1980s, researchers were encouraged to modify their experimental design strategies in order to reduce, refine, and also replace (3Rs) the conventional methods including animal experimentation. The 3Rs principles led to a dramatic decrease in the use of animals in research and development while also lowering the failure rate of pharmaceuticals [2].

When compared to in vivo studies, in vitro studies demonstrated time and financial savings, high yield, high reproducibility, and fewer ethical concerns [3]. As a result of their advancement and technological innovations, the microfluidic device was able to be created, opening up new possibilities, allowing the association of multiple components, and functioning as a “mini laboratory”, also known as a “lab-on-a-chip”, with possible application in areas such as chemistry, environment, bioenergetics and health [4,5,6]. 

The fabrication of microfluidic devices requires a set of procedures that enable the development of structures at a micrometric scale with great precision, in order to ensure a laminar flow of fluid in the microchannels [7]. The soft lithography technique is widely used for stamping or micromolding processes due to its ease, effectiveness, and low cost [8]. Complementary techniques, however, have been investigated for the creation of microdevices, which ensures a wider variety of possibilities for the employment of various polymers and structures. These techniques include photolithography and stereolithography [9].

The concentration gradient generator (CGG) is a type of microfluidic device capable of generating a concentration gradient via convection-mixing-based (tree-shape and altered-tree-shape), laminar-flow-diffusion-based (Y-shape), membrane-based, pressure-balance-based, droplet-based, and flow-based methods. All of these different techniques have been proposed and evaluated in a variety of experiments, allowing the study of numerous biological processes, such as cell migration, immunological response, wound healing, cancer invasion and metastasis, inflammation and chemotaxis, and the investigation of the concentration at which an element becomes harmful to an organism [10]. Compared to traditional macroscale evaluation methods, CGG microdevices allow for high analysis performance, with low reagent consumption, more efficient use of samples with limited volumes, a high surface-to-volume ratio, spatio-temporal resolution, portability and easy customization, control, and automation [11,12].

The CGG microdevice technology combines the advantages of microfluidics with a three-dimensional (3D) cell compartment that can preserve the biological complexity of cell models (3D cultures, including microenvironment or vascularization) and mimic drug evaluation, similar to animal models. Many drug candidates in different concentration ranges are evaluated at the same time, and different treatment regimens can also be explored using multiple drug gradient generators and parallel cell culture chambers [13]. 

The demand for novel medication development is at an all-time high, due to rising drug resistance and the emergence of new diseases, motivating the search for more efficient drug screening methods. The CGG microdevice approach to performing the antimicrobial susceptibility test (AST) is a simple, economic, and fast way to emulate a traditional AST and rapidly provide the minimal inhibitory concentration (MIC) of an antibiotic for a certain bacterial strain, at rates comparable to those of other miniaturized devices and automated AST instruments. The MIC value allows clinicians to prescribe appropriate dosages of the medication and stop bacteria from becoming resistant before being eradicated [14].

In this systematic review, our objective was to investigate how studies have applied micro-CGG for toxicological evaluation and for what purposes, in addition to the technological evolution in the development of these systems. The microfluidic device manufacturing and new technologies applied, the perspective of design and methodology of the CGG system, and the type of biological environment used to evaluate the substance’s toxicity, as well as the outcomes, were considered.

## 2. Materials and Methods

### 2.1. Search Strategy

We conducted a systematic search for articles that were published in the previous 10 years, including the years between 2011 and 2022. The articles selected, which are indexed in PubMed and Scopus, followed the Preferred Reporting Items for Systematic Reviews and Meta-Analyses (PRISMA) guidelines [15]. The criteria of interest selected were keywords in the following sequence: ((Concentration Gradient Generator) AND (Toxicity) AND (Microfluidic Device)), using the boolean operators (DecS/MeSH):

SCOPUS: (((TITLE (“organs-on-chips”) OR TITLE (“organs-on-a-chip”) OR TITLE (“microfluidic device”) OR TITLE (“lab-on-chips”) OR TITLE (microfluidics) OR TITLE-ABS-KEY (microfluidic)) AND PUBYEAR > 2010 AND PUBYEAR > 2010) AND ((SRCTITLE (toxicity) OR TITLE (toxicities) OR SRCTITLE (toxicological) OR TITLE (nanotoxicity)) AND PUBYEAR > 2010 AND PUBYEAR > 2010)) OR (((TITLE (“concentration gradient generator”) OR TITLE-ABS-KEY (“microfluidic gradient generator”)) AND PUBYEAR > 2010 AND PUBYEAR > 2010) AND ((TITLE (“organs-on-chips”) OR TITLE (“organs-on-a-chip”) OR TITLE (“microfluidic device”) OR TITLE (“lab-on-chips”) OR TITLE (microfluidics) OR TITLE-ABS-KEY (microfluidic)) AND PUBYEAR > 2010 AND PUBYEAR > 2010)) 

PubMed: (((TITLE (“concentration gradient generator”) OR TITLE-ABS-KEY (“microfluidic gradient generator”)) AND PUBYEAR > 2010 AND PUBYEAR > 2010) AND ((TITLE (“organs-on-chips”) OR TITLE (“organs-on-a-chip”) OR TITLE (“microfluidic device”) OR TITLE (“lab-on-chips”) OR TITLE (microfluidics) OR TITLE-ABS-KEY (microfluidic)) AND PUBYEAR > 2010 AND PUBYEAR > 2010)] OR (((((((“organs-on-chips”(Title)) OR (“organs-on-a-chip”(Title))) OR (“microfluidic device”(Title))) OR (“lab-on-chips”(Title))) OR (microfluidics(Title))) OR (microfluidic(Title)) AND (2011/1/1:2022/6/1(pdat))) AND ((“Concentration Gradient Generator”(Title/Abstract)) OR (“microfluidic gradient generator”(Title/Abstract)) AND (2011/1/1:2022/6/1(pdat))) Filters: from 1 January 2011 to 6 June 2022)

### 2.2. Selection Criteria

We only included original articles written in English published within the previous 10 years that used a microfluidic device capable of generating a gradient to analyze the toxicity of different concentrations of a substance to living organisms. The selection factors were in accordance with the PICO criterion we used: Problem: difficulty in generating a linear concentration gradient of a substance quickly and effectively; Intervention: use of microfluidics device to generate gradients; Comparison: to assess substances’ toxicity screening with concentrations generated by CGG and by pipetting; Outcome: toxicity assessment.

### 2.3. Exclusion Criteria

The following exclusion criteria were used: (i) reviews, (ii) publications written in languages other than English, (iii) indexed articles published in more than one database (duplicates), (iv) only microdevice fabrication protocols, (v) does not assess the toxicity effect in biologic organism, (vi) does not apply toxicology test, and (vii) the microdevice did not employ a concentration gradient generator.

### 2.4. Data Compilation

In this review, eight of the authors (N.M.E.V., M.P.N., A.H.A., L.D.R., J.B.M., F.A.O., C.S.L., A.T.L., M.N.P.C., and L.F.G.), in pairs, independently and randomly analyzed, reviewed, and assessed the eligibility of titles and abstracts according to the strategy of established search. The authors N.M.E.V., M.P.N., A.H.A., L.D.R., J.B.M., and L.F.G. selected the final articles by evaluating the texts that met the selection criteria. The authors N.M.E.V., L.D.R., J.B.M., F.A.O., C.S.L., and L.F.G. were responsible for the search for the characteristics and fabrication of the CGG with the collaboration and review of the authors A.T.L. and M.N.P.C. The authors N.M.E.V., M.P.N., A.H.A., L.D.R., and L.F.G. searched for the device microenvironment and toxicity techniques. All authors contributed to writing the entire text of this review.

### 2.5. Data Extraction

Four topics were used to analyze the papers under review, and they were represented in tables that addressed the following features: (1) characteristics, design, and fabrication of concentration gradient generator microfluidic devices for toxicity analyses; (2) microfluidic concentration gradient generators’ characteristics; (3) biological model used for toxicity evaluation; and (4) outcomes of the studies.

### 2.6. Risk of Bias Assessment

The selection of articles was performed in 2 pairs, and, in case of disagreement, an independent senior author decided on whether the article in question would be included. The data selected in the tables were divided by the authors into the groups already described above, and the checking of the data was carried out by the following group. In the case of disagreement, author L.F.G. made the final decision. 

### 2.7. Data Analysis

The data obtained in each of the tables were analyzed in percentages and range distribution to highlight the main characteristics, particularities, and exceptions, according to applicability.

## 3. Results

### 3.1. Selection Process of the Articles Identified According to the PRISMA Guidelines

We searched the PubMed and Scopus databases for publications from the last 10 years, considering the period from 2011 to March 2022 and following the selection inclusion and exclusion criteria already presented, resulting in 352 articles identified, comprising 254 articles from Scopus and 98 from PubMed. Of the 254 articles found in Scopus, 86 were excluded after screening because 31 were reviews, 43 were conference papers, 8 were book chapters, and 4 were not eligible. At screening, 56 articles from PubMed, comprising 47 duplicates and 9 reviews, were also excluded. Eligibility analysis was carried out following the selection criteria, and 139 of the 168 articles from Scopus (45 did not report the organism used in the study, 26 only reported the device development, 30 did not assess toxicity, 17 did not apply the concentration gradient generation in the device, and in 21 the toxicity was assessed outside the device) and 38 from PubMed (8 did not report the organism used in the study, 16 only reported the device development, and 14 did not assess toxicity) were excluded. Thus, only 33 unduplicated full-text articles [16,17,18,19,20,21,22,23,24,25,26,27,28,29,30,31,32,33,34,35,36,37,38,39,40,41,42,43,44,45,46,47,48] were included in this systematic review, 29 from Scopus and 4 from PubMed, as shown in Figure 1.

The 33 selected studies were analyzed regarding the microdevice fabrication, the CGG characteristics, the biological model, and the main outcomes. Due to the different biological approaches of the selected studies, the tables were organized internally by the four main types of organisms used for toxicity analysis inside of the device: 7 studies used microorganisms (bacteria and nematodes) (21%) [16,17,18,19,20,21,22], 5 used microalgae (15%) [23,24,25,26,27], 19 used tumor cells and other models (58%) [28,29,30,31,32,33,34,35,36,37,38,39,40,41,42,43,44,45,46], and 2 used zebrafish embryos (6%) [47,48]. 

### 3.2. Characteristics, Design, and Fabrication of Concentration Gradient Generator Microfluidic Devices for Toxicity Analyses

The technologies utilized in the design, manufacturing, finishing, and innovations of microfluidic devices to generate concentration gradients for toxicological analysis and drug screening in the studies included in this systematic review are summarized in Table 1. General analysis was initially performed regarding the date of publication of the 33 articles with the division performed according to the organisms used for the toxicity assessments (microorganisms, microalgae, tumor cells and other models, or zebrafish embryos), pointing out that in the studies carried out in the last 5 years [16,17,18,19,23,24,28,29,30,31,32,33,34], the use of microorganisms and tumor cells and other models was more prominent, showing a higher incidence and demand for microdevices focused on efficient drug screening. This specific division by the biological model of Table 1 did not necessarily have a connection with particularities in the manufacturing techniques related to the microfluidic characteristics.

Regarding microfluidic device fabrication, all evaluated devices were manufactured in-house, and the methods used in the studies are organized and described in this paragraph. Among the materials used, polydimethylsiloxane (PDMS) was predominant (91%) [16,17,19,20,21,22,23,24,25,26,27,29,30,31,32,33,34,35,36,37,38,39,40,41,42,43,44,46,47], and the device manufacturing technology used with this material was soft lithography, totaling 91% of the studies [16,17,19,20,21,22,23,24,25,26,27,29,30,31,32,33,34,35,36,37,38,39,40,41,42,43,44,46,47], being, in the studies that used microalgae [23,24,25,26,27], reported in 100% of the cases. Only 9% of the studies reported other technologies and materials [18,28,48], such as ultraviolet (UV) photolithography (3%) [48] in glass applied in the Zebrafish embryo model, silicon micromachining (3%) [28] with silicon in the tumor cells and other models groups, and 3D printing using a polymer as the main material mold, which was applied in one study from the microorganisms group [18].

Soft lithography methodology consists in pouring a polymer over a mold. The mold fabrication was performed mainly by UV photolithography (72.2%) [16,17,20,21,23,24,25,26,27,30,31,33,34,35,36,37,38,39,42,43,44,46,48], and 58.3% [16,17,20,21,22,23,24,25,26,27,31,33,34,35,36,37,38,39,41,42,44,48] of the materials used were negative photoresists, such as SU-8 and S1800, while only 5.5% [39,45] of studies utilized positive photoresists, such as AZ, all of these being from the tumor cells and other models group. Once again, only microalgae studies were unanimous on mold fabrication, using SU-8 material. A total of 15% of studies reported other mold manufacturing methods [19,22,29,41,47], with 12% reporting the use of computer numerical control (CNC), which was applied in different types of biological models [19,22,41,47], and 3% reporting the use of laser-based technology, which was applied in one study that used tumor cells and others as a model [29]. In 27.7% of studies [19,22,29,41,47], other materials were used, such as mold (glass, silicon, PDMS, poly(methyl methacrylate) (PMMA), copper, and Pro/Cap50), and only one study, which used a tumor human cell model, did not report this information [32]. 

Most of the devices (73%) were reported to have more than one sandwiched layer [16,17,18,19,22,23,25,26,27,29,30,31,32,33,34,35,36,37,38,39,43,44,46,47,48], 18% used structures in only one layer (two microalgae studies [20,24] and four cell studies [28,39,41,42]), 6% did not report the layers used [21,45], and in the study on the microdevice manufactured by 3D printing, this information was not applicable [18]. After the device’s completion, the channels were commonly sealed with a glass cover (62.8%) [16,17,21,23,25,28,29,30,31,32,33,35,37,38,39,42,43,44,46,47,48] or with polymeric sealing (31.4%) [19,20,22,24,26,27,34,36,39,41]. Only the studies with the zebrafish embryo model had unanimity in glass cover, and in 5.8% [18,45], this was not applicable. The bonding techniques used to cover the microdevices were, mainly, plasma bonding (63.6%) [16,20,22,23,25,27,29,30,31,32,34,35,37,38,39,41,43,45,46,47], followed by uncured PDMS (12.1%) [17,19,26,42] and anodically bonding (6%) [28,48]. A few studies reported other techniques (6%) [36,39,44], and in others, this information was not applicable or reported (12.3%) [18,21,24,33]. The sealing techniques were evenly split between all four groups of the biological model. 

New technologies analysis showed innovations in their fabrication or materials, such as the development of facilitators in relation to the main reported problems in microfluidics, such as shear and bubbles (27%) [16,17,21,27,35,37,38,42,44], cost reduction, and rapid prototyping (12%) [18,28,36,48], and integration of other systems (9%) [19,20,24], such as electrodes, for example. Another 24% [22,23,29,31,32,34,41,47] presented precise technologies for the analysis of the organism in question, enabling customized development. A total of 27% of articles did not report new technologies [25,26,30,33,39,43,45,46]. Finally, regarding the dimensions of microfluidic devices used in the selected studies, mainly in the culture chamber and channel parts, the measurements were very particular for the purpose of the study; the larger chambers used a higher concentration of microorganisms in cultures or 3D culture. The devices made for the microalgae model had measurements with less variation, and, for the zebrafish embryos model, the chamber height was much shorter than for the others. According to the design and the structures of the microdevice developed by each author, some of the CGG’s particularities are exemplified in Figure 2.

### 3.3. Concentration Gradient Generator Characteristics of Microfluidic Device

For the development of microfluidic devices capable of generating concentration gradients, it is necessary to establish general and functional characteristics, such as, as analyzed in Table 2, the methods and types of systems used to generate gradients, number of concentrations generated, linearity of the gradient, variation in concentrations of the compounds evaluated, and time to achieve gradient stability, as well as the duration of stability, in addition to information on simulation methods, validation, and advantages. Most studies used the Christmas tree gradient generation system (Figure 2A–D,F,G,I) [21,22,23,24,25,26,27,29,30,34,35,36,37,38,39,42,43,44,45,46,47,48], which uses the convective method, either associated or not associated with other systems or with certain modifications. Less frequently, other systems also used convective methods, such as T-shaped channels (6%) (Figure 2G) [20,37], serpentine channels (Figure 2A–D,F–I) [23,25,34,37,40,41,42,44] cascade mixing (Figure 2H) [40], and 3D microchannel networks [18] (3% each). Diffusion methods used to generate gradients were associated with Y-junction systems (6%) [21,28], the snake model [24], droplet generation (Figure 2E) [16], static-pressure-driven CGG [31], and membrane systems (Figure 2J) [17], with 3% each. 

Interestingly, two of the studies, one from the microorganisms group and another from the tumor cells group, used a combination of convective and diffusive methods, such as serpentine/T-shaped channels [20], and Christmas tree/Y junction (Figure 2G) [37]. Only three studies did not report the generation method used, and the systems used were centrifugal CGG [19] and circular concentration gradient [33], while the study by Qin, Y.X. et al. reported neither the method nor the system [32]. The CGG structure was designed according to the gradient system used and the generation method, to ensure efficiency in toxicity screening. These important aspects of the CGG structure are highlighted in Table 2.

An alternative way to evaluate the functionality of the structures and the efficiency of the generation of gradients is the use of software to simulate the flow. In this review, only 18% of studies used COMSOL (software for multiphysics simulation) [17,18,28,30,31,41], mainly those on tumor cells and other models and microorganisms. The study by Han, B. et al. (microalgae group) performed the simulation through computational fluid dynamics (CFD) [24]. In the studies with embryos, simulation was not reported [47,48].

The devices developed in the selected studies generated from 2 to 65 different concentrations, with the greatest variation observed in the tumor cells and other models group, at 3 to 65 concentrations generated, and the lowest in the microalgae group, at 5 to 8 concentrations, with the microorganism and the zebrafish embryo groups having produced from 5 to 8 and 2 to 24 concentrations, respectively. The generated concentration values were reported in 91% of the studies, and they were considered linear [16,17,19,20,21,22,23,24,25,26,27,28,29,30,31,34,35,36,37,38,39,40,41,42,43,44,45,46,47,48]. 

For the creation of the perfect gradient, a certain amount of time is required, and only 30% of the studies reported these data, varying from 1 to 1800 s [16,26,27,28,36,37,41,42,45,48]. The stability time, also an important parameter, was reported in only 12% of studies [25,26,27], varying between 20 and 50 min in the microalgae group, with one study from the tumor cells and others group reporting an indefinite time of maintenance [28], while the other groups did not report this parameter.

One of the final steps in the development of the devices, validation, which aims to ensure the correct functioning of the gradient generator system, was reported in 55% of studies, with the main method used for this purpose being fluorescent agents [18,20,21,22,25,26,27,28,31,33,34,35,37,38,42,43,45,47], followed by food coloring, used in 12% of studies, one from the microorganism group [19] and three from the cells group [29,36,40]. Only one study (3%) used Doxorubicin (DOX) for validation [30]; the other 30% did not report this step [16,17,23,24,32,40,41,44,48]. 

Regarding the concentrations of drugs or stimuli used in the study of toxicity, some of the more frequently used substances showed a similar pattern. For the antibiotic toxicity screening in microorganisms, the most frequently used antibiotics were Ampicillin (AMP) and Tetracycline (TAC), with concentrations ranging from 0 to 13.1 μg/mL [16,18,19], while the concentration of Ciprofloxacin (CIPRO) ranged from 0 to 96 μg/mL [17]. For the evaluation of the toxicity in microalgae, the range of concentrations of copper (II) sulfate varied from 0 to 4.375 μM, and for mercury (II) chloride, from 0 to 4 μM [24,25]. The concentration variation in the main chemotherapies applied in toxicity screening in the tumor cells group varied from 0 to 600 mg/mL for 5-Fluorouracil (5-FU) [41,43,45,46], from 0 to 400 mg/mL for Cisplatin (CDDP) [29,35,45], 0 to 3.4 mg/mL for Paclitaxel (PTX) [35,37], and from 0 to 0.01825 mg/mL for DOX [30,36]. For studies related to embryogenesis, a lower range was used (0–100 μg/mL) for Adriamycin (ADM), DOX, 5-FU, and CDDP [48].

Some advantages regarding the CGG system and microdevice structure were reported in 55% of the studies. Shear-free fluid flow was a concern considered by 32% of the works, where shear minimization was provided, mainly, by the shape of the mixing channels, difference in heights in relation to the culture chamber, and use of splitting–mixing systems associated with serpentine channels [18,19,28,30,36,38]. Automation was also considered by 26% of the studies, so vacuum pressure channels (Figure 2A), centrifugal force [19], snake-channel torque-operated valves [26], and centripetal geometry [47] were used to minimize handling and optimize the generation of gradients [19,24,26,44,47]. To guarantee the linearity of the gradients, 21% of the studies reported the optimization of the structures by modifying the length of the channels (Figure 2A,H) [40,44], using micropillars in the culture chambers [30], radial splitting–mixing integration with a serpentine channel [33,45], and cascaded mixing (Figure 2H) [23,30,33,40]. In addition, 11% reported a concern regarding the high performance of these devices, the number of concentrations generated using radial splitting–mixing integration with a serpentine channel, and centripetal geometry together with the arrangement of concentric serpentine channels [45,47]. Only 5% of the studies reported a concern with mimicking the gradient in vivo [28] and reusing the developed devices [43].

### 3.4. Biological Model Used for Toxicity Evaluation in the CGG Microfluidic Device

Table 3 shows the details of the biological model used, the characteristics of the culture environment, and the toxicity conditions analyzed. The main microorganisms used as biological models were, firstly, bacteria (71.4%), with the great majority of the studies choosing different *E. coli strains* [16,17,18,19]—with exception of the study by DiCicco [21], in which a canine bacteria species (*S. pseudintermedius*) was used—and, secondly, Caenorhabditis elegans (*C. elegans)* nematode (28.6%) [20,22]. Both models were utilized for antibiotic toxicity screening—with exception of the study by Zhang B [20], which employed manganese chloride combined with vitamin E, resveratrol, and other substances. The most tested antibiotics were AMP [16,19], CIPRO [17,18], and TAC [16,18], followed by Kanamycin (KAN) [16], Amikacin (AMK) [18], Fosfomycin (FO) [21], and Amoxicillin (AMX) [22], with an incubation time between 4 and 72 h. The longer periods of incubation were associated with the evaluation of genetic mutation and antibiotic resistance. The drugs’ flow rates were reported in less than half of the studies (42.9%) [20,21,22], with a range of 10 to 300 μL/h, and the organisms were mostly cultured intra-CGG (85.7%) [16,17,19,20,21,22] and in 2D culture, with only two studies reporting the use of 3D culture (28.6%) [16,17], one being a co-culture. The average number of organisms for the studies that used bacteria was around 10^6^ cfu/mL (10^8^ for canine bacterium) and, for those based on nematodes, 1 worm/mL. Regarding the culture environment, the principal medium employed for bacteria culture was Luria–Bertani broth (for the *E. coli strains*), excluding the study based on *S. pseudintermedius*, which used Columbia agar associated with Tryptic soy broth plus glucose, and for *C.* elegans, a nematode growth medium was chosen. All bacteria were incubated with temperatures ranging from 30 to 37 °C, and the nematode studies applied lower temperatures around 20 to 25 °C. 

Among the marine microalgae studied, 80% were *Chlorophyta* (green microalgae) [23,24,25,26,27], the most frequently seen species being *P. subcordilformis* (33.3%) [24,25,26,27], *P. Helgolandica var. tsingtaoensis* (25%) [25,26,27], and *Chlorella* sp. (16.7%) [23,26], all of which are from the previously cited phylum. This model was utilized for evaluating the toxicity of water pollutants, mainly metals and composts, most frequently copper (80%) [24,25,26,27], followed by mercury [24,25], cadmium [24,27], lead [25], and zinc [24], as well as other substances, such as sodium hypochlorite [23] and phenol [27]. The flow rates of the pollutant solutions and the exposure times in the toxicity evaluations varied greatly, from 0.1 to 50 μL/min and 1 to 72 h, respectively. These toxicity assays were mainly conducted intra-CGG (80%) [23,24,25,27], in an F/2 medium (80%) [24,25,26,27] (an enriched seawater medium was used in one study [23]), in a 2D arrangement, with an average amount of microalgae of 10^5^ individuals or a range between 240 and 580 cells/μL, maintained mainly at 25 °C and in controlled light illumination of 60 μmol photon/m^2^/s.

Most of the selected studies used human cells (87.1%) [29,30,32,34,35,36,37,38,39,41,43,44,45,46] as the biological model for chemotherapy toxicity screening, consisting, basically, of different types of carcinoma (77.8%) [30,34,35,36,37,38,39,41,43,44,45,46], with the exception of kidney [29], endothelial [35], bronchial epithelial [32], and fibroblast cells [37], which were not necessarily used for the testing of anticancer drugs. Five studies opted for the use of cells from other organisms, such as embryonic stem cells from mice (9.7%) [28,31,43], insulinoma cells from rats (3.2%) [33], and saccharomyces yeast cells (3.2%) [42]. The employed test substances were, mostly, anticancer drugs, comprising 5-FU (26.3%) [35,41,43,45,46], CDDP (21.1%) [29,35,44,45], PTX (15.8%) [35,37,39], DOX (10.5%) [30,36], and, in lower frequency, Rapamycin [28], Gentamycin (GM) [29], Cyclosporin (CsA) [29], Cimetidine (Cim) [29], Irinotecan [34], Acetaminophen (APAP) [38], Pyocyanin (PCN) [39], and Cyclo-phosphamide (CP) [45], used at a percentage of 5.3% each, with the exception of cigarette smoke extract (10.5%) [32,39], hydrogen peroxide [31], glucose associated with glipizide [33], ascorbic acid [42], the combination of galactose, raffinose, and iron (III) chloride [42], and Celecoxib [43] (5.3% each), and their flow rates (average of 3.4 μL/min) and time of exposure (from 2 to 168 h) were extremely varied. Only 10.5% of the selected studies cultured the cells’ extra CGG system (Figure 2C) [34,35], that is, in a different layer from the one used for the generation of gradient concentrations or outside microfluidic devices, and the majority applied 2D culture (52.6%) (Figure 2A,I) [31,32,38,39,42,43,44,45,46], followed by 3D co-culture (15.8%) (Figure 2G) [29,30,37], spheroids (10.5%) (Figure 2C,D) [34,41], 3D culture (10.5%) [33,36], both 2D culture and spheroids (5.3%) [35], and both 2D and 3D cultures (5.3%) [28], using from 104 to 2.5 × 10^7^ cells/mL dispersed mainly in Dulbecco’s modified Eagle medium (DMEM) and its variations (42.1%) [29,30,31,35,36,38,44,45], followed by Roswell Park Memorial Institute (RPMI-1640) medium [32,33,37,41,44,46], and Eagle’s minimum essential medium (EMEM) [39,41]. All cells were incubated at 37 °C (aside from *Saccharomyces* [42], which were cultured at 30 °C), with a 5% CO_2_ humidified atmosphere. 

Only two studies used the zebrafish embryos as the biological model [47,48], culturing them in 2D arrangement and intra-CGG, but with different purposes. One of the studies [47], which was performed with 10 to 12 eggs per chamber, focused on lead acetate and copper sulfate toxicity screening by exposing the embryos to these pollutants for 48 h, using flow rates from 5 to 30 μL/min and incubating them in an aerated ultrapure water medium supplemented with nitric acid and sodium hydroxide at 28.5 °C. The second study [48] was performed with one embryo/chamber for chemotherapy toxicity assessment, the drugs employed being ADM, DOX, 5-FU, and CDDP, as well as vitamin C, in different stages of embryo development (4 to 72 h post-fertilization), with the flow rate of 4 μL/min, with incubation in an E3 embryo medium at 26 °C, alternating between anoxia and normoxia.

### 3.5. Toxicity Screening Evaluation and Outcome of the CGG Microfluidic Device

Table 4 highlights the main points of the proposal, evaluation, and outcome of the selected studies. The main proposal of studies on the microfluidic devices that used microorganisms (bacteria and nematodes) as a model was to perform an AST with single or combined (due to the antagonism or synergism effect) drugs with different exposure times, using the MIC value as a reference to compare the results with the gold-standard method, searching for the best efficiency while using the lowest amount of drug possible, as assessed by cell viability fluorescent techniques, as well as the influence of the drug’s concentration on genetic alterations and mutations that lead to drug resistance, an extremely relevant issue nowadays due to widespread misusage of antibiotics. The nematodes were used for different purposes; one study [20] evaluated the behavioral response of the worms by fluorescence imaging in the face of manganese toxicity and the protective effect of natural antioxidants while the other [22] evaluated the effectiveness of certain antibiotics on the treatment of bacterial infection on nematodes, either associated or not associated with natural substances, showing these to be of value when treating the infection. 

For marine microalgae, the main concern was the toxicity of chemicals linked to environmental contamination, either individually or in combination. These compounds were assessed using viability and motility techniques, which revealed varying sensitivities between different phytoplankton species. One study [23] concluded that *Chrorella* is more resistant than *Pyraminmonas* sp. to NaClO and the other microalgae, indicating the greater resistance of *P. subcordiformis* and *P. helgolandica* var. *tsingtoaensis* to all metals tested, especially CuSO_4_, which was shown to be the most toxic. 

Most studies that proposed the screening of antitumor substances in cells from humans and animals assessed its efficiency via the use of different fluorescent dyes associated with cell viability evaluation (Calcein AM/Pi, Hoechst 33342, and Annexin-V-FITC), concluding that the drugs have a time/dose-dependent effect in almost all cases in which a drug was tested singly, and, also that the combination of drugs had better efficacy in lower dosages, with similar results seen for assays performed on Petri dish cultures. Only a few studies [28,29] assessed the effects of chemotherapy (or, in two studies [32,40], cigarette smoke extract) on normal cells, through the evaluation of apoptosis or oxidative stress via a reactive oxygen species (ROS) assay, showing that the toxicity and the malignant transformation of cells depend on the time of exposure. The study by Fernandes [42] was the exception in these approaches, evaluating the α-synuclein (aSyn) production and aggregation in Saccharomyces cerevisiae exposed to iron and ascorbic acid, due to the supposed protective effect of these substances.

Each of the zebrafish embryo investigations had a different objective. By using morphometric and behavioral analysis, one study [47] showed the damage effects of metal in different stages of embryo development. The second study [48] focused on the effects of chemotherapeutics normally used in embryo development and the ability of vitamin C to reduce harm.

As for the advantages of utilizing microfluidic devices rather than traditional macroscale methods, the great majority of the studies reported similar benefits. Firstly, some of the studies reported that the results obtained with microdevices correlate very well with those obtained using conventional methods, sometimes even mimicking more accurately in vivo conditions, showing that the technology in question can be adequately applied when studying toxicity. With that in mind, one of the most important aspects reported is the possibility provided by microfluidic CGG’s ability to generate a very sizeable number of different concentrations (up to 65 in the studies analyzed) in a single device and, consequently, the possibility of conducting multiple parallel assays, both of which, allied with the prospect of automation of processes (such as the generation of the concentration gradient and metabolite collection), can significantly diminish the time expended and make this a high throughput method for toxicity screening. Other very significant advantages brought about by this technology include the small size of the devices, which translates to less space occupied, making it possible to have multiple devices running multiple assays at the same time, further increasing the throughput, and also, the low quantity of reagents used, decreasing the cost of the tests. Besides that, the microdevices can be easily and rapidly fabricated, with different well-established fabrication processes and a variety of materials, and easily operated, as well as integrated with other traditional techniques, combining the advantages of both. Microdevices are also more preferable for 3D cultures than some of the more traditional methods, and they make it possible to create microenvironments that are more like those seen in nature, producing results that are more trustworthy. The analysis of microfluidic devices can also be carried out via a variety of methods, providing the researcher with a lot of design flexibility. The device’s versatility, which allows it to be developed in an endless number of ways for various purposes, with various test chemicals and biological models in mind, is still another significant advantage that can be seen. All these parameters are described in Table 4.

According to their toxicity methodologies and the biological models employed for this testing, the studies’ findings are described in Figure 3 in conjunction with the major features that are considered in this systematic review.

## 4. Discussion

Advances in microfluidic device development technology for toxicity screening have provided remarkable advantages over conventional two-dimensional cultures due to the reduction in the sample consumption, reaction time, and cost of the operation. This systematic review gave a broad overview of the main aspects and trends regarding the manufacture of microfluidic devices, the promotion of the CGG’s development to boost the effectiveness of its chemical and drug toxicity screening, and the most tried-and-true biological models for addressing issues concerning environment and medical treatments.

Regarding microfluidic device fabrication, all were manufactured in-house, providing device customization for more efficient testing, which was specific to each biological model used. There is still a strong tendency to use materials and manufacturing techniques such as PDMS and soft lithography (91%), but recent articles search for more sophisticated technology, such as a 3D printing, silicon micromachining, and direct writing photolithography using glass. PDMS is the most commonly used material in microfluidics, because of its flexibility, biocompatibility, nontoxicity, good stability, and high transparency [49], even though earlier studies—some from more than a decade ago—brought up disadvantages, such as the absorption of small molecules [50], its incompatibility with organic solvents [51], and its vapor permeability [52], and more recent articles have questioned its practicality and widespread use, citing the difficulty of translating results obtained with it to other materials and its poor scalability for commercial purposes [53,54,55,56] as major concerns. The studies that did not use PDMS reported the use of materials such as silicon and glass which have, roughly, the same advantages as PDMS beside hydrophilic capabilities, reusability, and flexibility [45]. The biomedical field finds 3D printing to be a highly valuable technology for diagnostic and/or therapeutic purposes; its applications range from tissue engineering to microscale robotics and biosensors, besides rapid prototyping flexibility and a variety of forms and functions, having the advantages of precisely controlling the spatial distribution layer-by-layer, the generation of heterogeneous microorgans, and 3D cellular arrangement on a chip [57,58]. Only one study [39] utilized thermoplastics in some way, which is an interesting finding, given that, in recent years, materials such as polycarbonate (PC), poly (methyl methacrylate) (PMMA), and cyclic olefin copolymer (COC) have been gaining notoriety and have been widely used in industry when aiming for the fabrication of a product [53,54]

The studies that used conventional manufacturing varied the type and number of molds. Photolithography was the most used method of fabrication (79%), mostly due to its high accuracy, despite its high cost [59]. A study on optimization of SU-8 microstructure in high-transparency masks, printed in a photomask, however, showed the possibility of their fabrication with a low-cost process and without the requirement of cleanroom facilities [60]. Laser cutting techniques, as well as CNC, when compared to traditional photolithography and etching methods, have the advantages of being a simple, fast, and direct-write process for the fabrication of different geometrical shapes. Both techniques provide complex geometries with different layers, normally more than one layer (72.3%), with a micrometer scale. The layers represented the different environments and testing functions of the microdevice, providing greater efficiency within its complexity. 

Most studies (73%) reported the use of new technologies, aiming to minimize microfluidic problems and also to innovate in the material and manufacturing of microdevices [16,17,18,19,20,21,22,23,24,27,28,29,31,32,34,35,36,37,38,41,42,44,47,48]. In addition, some studies proposed technological advances integrating electronic systems (9%) [19,20,31]. The evolution in the fabrication of complex and adaptive microfluidic devices was evidenced in the selected studies with implementations that showed significant advantages of the CGG used, such as its ability to create sophisticated and precisely defined gradient profiles. 

The CGG system is a faster and more accurate method for drug and chemical pollutant toxicity analysis. It only needs a small amount of reagent for multiplex analysis, which lowers the cost. It is also capable of screening at the molecular and cellular levels and has multistep liquid-handling capabilities, which is especially useful for complicated screening procedures, in addition to its features of miniaturization, integration, and automation of analytical systems [36,61]. 

The method of gradient generation was based on two patterns, convective and diffusive. Most of the selected studies used the convective method for gradient generation (67%), which is a simpler and easier method for drawing and calculating. In convection-based gradient generators, the concentration gradient depends on the flow field, which can produce shear stress above the physiological limit endured by cells. The diffusion-based gradient generator, on the other hand, offers isolated chambers due to the interface, and the inside reagents are protected from the outside shearing [10]. The Christmas tree generation system was the most used (61%), associated and not associated with other systems, which indicates the frequency of the convective pattern, and its main advantages were its simple design and a well-defined concentration range, allowing isolated assessment of each concentration. However, this pattern can be integrated with other systems such as Y-junction systems or in two separate layers, one in which the convection pattern is evident (the CGG layer), forming the concentrations, and one containing the culture chamber, in which the different concentrations flow through diffusion. A few studies used similar systems to the Christmas tree, such as serpentine channels, cascading mixing, and T-shaped systems, which have certain advantages, for instance, fewer stages [62]. The studies that used the diffusion pattern applied a variety of gradient systems, such as Y-junction, membrane, and droplet generation. This last system shows difficulty in controlling flow and concentration while maintaining the droplet shape, two crucial parameters for toxicity assessment.

Gradient linearity is the expected behavior of CGG, being reported in 91% of studies due to the need to assess dose dependency on drugs and toxicity. The studies used two methods to analyze CGG linearity and performance: flow simulation and validation. The flow simulation occurs in a stage before the CGG manufacturing, allowing quick design adjustment, but only seven studies (21%) reported analysis using the COMSOL software, likely due to its high cost and requirement of an expert user, making access to it difficult. CGG validation is a different type of analysis that can only be performed once the microdevice is complete. The most employed substances for this method were fluorescent agents (55%) [18,20,21,22,25,26,27,28,31,34,35,37,38,42,43,45,47] followed by dye solutions (12%), bringing a visual analysis of the flows of the channels and the concentrations generated [19,29,36,38,39]. Some studies also performed a quantitative analysis to be compared with the final concentrations. 

Considering the publication year of the 33 studies included in this systematic review and the different approaches for toxicity screening, the studies from the first five years directed greater attention to environmental problems, such as contamination of the seas by metals and other pollutants (60%) [25,26,27,35] and advances in treatment with chemotherapeutics, seeking better drug combinations for better efficiency (63%) [35,36,37,38,39,40,41,42,43,44,45,46], both contexts being analyzed in studies using zebrafish embryos [47,48]. Currently, studies are more focused on effective antitumor therapies (37%) [28,29,30,31,32,33,34] and pollutants’ toxicity in the marine microenvironment (40%) [23,24], while the concern over antimicrobial treatments has grown (43% [20,21,22] to 57% [16,17,18,19]). 

Among the most tested antibiotics in the toxicity analysis, there was a slight predominance of Ampicillin and Ciprofloxacin, which are effective against a wide range of both Gram-positive and Gram-negative bacteria, while having distinct modes of action [63]. The concentration range of both drugs showed a similar pattern (from 2 to 16), showing MIC values consistent with the gold standard of conventional analysis, being more efficient in terms of analysis time and material consumption, and allowing combined-drug analysis for synergism and antagonism effects, using a drug exposure time from 4 to 72 h. The use of prolonged subtherapeutic levels is a concern regarding bacterial resistance, with microfluidic devices having been shown to be more efficient for this analysis than conventional techniques due to the possibility of mimicking the in vivo microenvironment, while guaranteeing high performance [64]. The main bacteria utilized as a model was the Gram-negative Escherichia coli (*E. Coli)* (for example, *E. Coli* k-12), which may cause severe food poisoning and is a global health problem due to the rise in antibiotic resistance. Due to its unrivaled fast growth kinetics, high-cell-density cultures, and quick and simple exogenous DNA transformation, this species of bacteria is the most popular for use in toxicity assays [65]. Almost all bacteria were cultured inside the CGG system in a 2D culture dispersed in a medium from 30 to 37 °C, with the exception of the study by Sweet [18], which cultured them in a separate system (extra CGG), and the studies by Zeng [16], which used 3D culture, and Nagy, who used 3D co-culture [17].

A few studies also used nematode *C. elegans* as a model for toxicity screening, assessing the influence of antibacterial activity with various rhubarb components [22], and dopaminergic neurotoxicity induced or not by manganese associated with antioxidant elements [20]. This is a strong model organism because of its small size, optical transparency, short life cycle, and genetic tractability, among other advantageous traits, such as the ability to be infected by a variety of human pathogens and low cost of maintenance [66]. This microorganism was also cultured inside the CGG system in 2D culture dispersed in the medium from 20 to 25 °C. 

Through the evaluation of metals and contaminants, a number of research articles have addressed the problem of environmental toxicity. Global pollutants such as mercury and lead, for instance, have an impact on both human health and the ecology around the world [67]. Microalgae have reportedly been used for biological detoxification, effluent treatment, control of toxic metals in natural waters or effluents, and control of toxic metals in naturally or industrially contaminated waters [68], as well as to retain and immobilize some compounds. Thus, it is essential to create tools that can investigate and aid in the creation of new technologies that are beneficial for the environment and, by extension, for human health and quality of life. Although other metals, such as arsenic (As), cadmium (Cd), chromium (Cr), lead (Pb), and mercury (Hg), are poisonous to microalgae, they can ingest trace amounts of metals, including boron (B), cobalt (Co), copper (Cu), iron (Fe), molybdenum (Mo), manganese (Mn), and zinc (Zn). Low-hazardous metal and compost concentrations can promote the growth and metabolism of microalgae because of the hormesis phenomena [69]. 

The metals Cu [24,25,26,27], Cd [24,27], and Hg [20,24] were evaluated the most often, singly or combined with other chemical elements, likely due to the high plastic accumulation in oceans from unrecycled waste and its decomposition [70] or the increase in mining and industrial activity, leading to mercury deposit [70]. The concentration range of these metals was similar (from 0 to 4 µM), varying from five to eight different concentrations tested. Green microalgae (*Chlorophyta*), the main model used, are photosynthetic protists and one of the groups of algae most closely related to terrestrial plants, also being used as indicators of water quality and having significant ecological importance, as they are components of phytoplankton, one of the primary producers in the food chain [71]. The microalgae were cultured mainly inside the CGG system—with the exception of the study by Zheng [26], which cultured them outside the CGG system—in 2D culture dispersed in the F/2 medium at about 25 °C, in controlled light illumination of 60 μmol photon/m^2^/s, close to normal environmental conditions.

Zebrafish embryos are frequently used in metal toxicity studies due to their ability to grow outward and having clear enough bodies to be examined under a standard optical microscope [72]. One study in this review evaluated the Pb and Cu toxicity, singly and combined, regarding its teratogenicity in different stages of embryo development, such as the larval, juvenile, and adult stages [47]. Another study used this model to analyze different types of chemotherapy drugs and the protective effect of vitamin C during treatment, evaluating their influence according to the developmental embryo stages [48], using drug doses significantly lower than those applied in the tumor cell and other models group. These embryos were cultured inside the CGG system in 2D culture dispersed in different medium conditions at about 27 °C, and the toxicity evaluation occurred from 1 to 72 h of exposure.

The vast majority of the tumor cell and other model group studies performed antitumor toxicity screening using various tumor cell types from human [26,30,31,32,33,34,35,36,37,39,40,41,42] or animals [33] and normal human cells [29,32,35], evaluating, primarily, the effect of the drugs 5-FU [35,41,43,45,46], CDDP [29,35,44,45], PTX [35,37,39], DOX [30,36], isolated or combined with others. CDDP and 5-FU combined are considered the standard antitumor treatment, and PTX followed by CDDP showed greater antitumor activity [73]. The toxicity of isolated Doxorubicin occur via acting on DNA by slowing or stopping the proliferation of cancer cells by inhibiting an enzyme called topoisomerase 2, their cardiotoxicity is the main factor restricting its use, and the total cumulative dose is the only factor currently utilized to predict toxicity, with microfluidics providing a new form of assessment [74]. 5-FU and CDDP also have activity on DNA, inhibiting thymidylate synthase, and crosslinking with the urine bases on the DNA to form DNA adducts, preventing repair of the DNA, leading to DNA damage and subsequently inducing apoptosis within cancer cells, respectively. The dose of these drugs was significant compared to other drugs, ranging from 0 to 600 mg/mL for 5-FU and 0 to 400 mg/mL for CDDP. PTX has a different antitumor mechanism, promoting the assembly of tubulin into microtubules and preventing the dissociation of microtubules, blocking cell cycle progression, preventing mitosis, and inhibiting the growth of cancer cells, being used, in the studies, in lower doses, from 0 to 3.4 mg/mL [75]. 

These cells were mainly cultured inside the CGG system in 2D conventional culture dispersed in DMEM or RPMI media, being mostly supplemented with fetal bovine serum (FBS) and other supplements at 37 °C. A few studies also used 3D culture and co-culture with different tumor or normal cells, and some studies specified the use of spheroids, a variation of conventional 3D culture. 2D cell culture models have been used to assess the toxicity or efficacy of drug candidates due to the ability to anticipate drug responses, but they have been found to be comparatively weak in comparison to 3D cell cultures, which have better functional and phenotypic characteristics, as well as predictability of therapeutic effectiveness [34,35]. In vivo, cells are arranged spatially into three-dimensional (3D) patterns that are encircled by an extracellular matrix (ECM), which leads to cancer cells growing in 3D cultures; in comparison to 2D cultures, these cells are more resistant to cytotoxic drugs [76]. Spheroids are one of the most relevant and modern models for cancer research. Their morphology and physiology are similar to those of a tumor in vivo, showing a network of cell–cell interactions, a 3D structure, the presence of a natural extracellular matrix, and nutrients, metabolites, and oxygen gradients [77,78]. 

Other contexts also used normal cells without the influence of chemotherapeutics. Two studies evaluated the influence of dose and time of exposure to tobacco extract on the malignant transformation of normal bronchial cells. The tobacco epidemic is one of the biggest public health threats the world has ever faced; there is no safe level of exposure to tobacco, and chronic cigarette-smoke-induced time-dependent epigenetic alterations can sensitize human bronchial epithelial cells for transformation by a single oncogene [79]. The study by Fernandes [42] investigated the basic molecular effects of aSyn in the context of living cells, with human aSyn being expressed in yeast and found to induce dose-dependent cytotoxicity, while iron (III) chloride and ascorbic acid were shown to have a protective effect [42]. The molecular basis of various human diseases has been extensively researched using Saccharomyces cerevisiae as a model organism. It is most well-studied in eukaryotic cells, while also being the easiest organism to grow under controlled circumstances and to manipulate genetically [80]. The study by Luo [33] used the INS-1 cells, which are a widely used and well-established model for the study of diabetes and their property of glucose-stimulated insulin secretion [81]. 

A relevant aspect in studies with microfluidics devices evaluated in the biological model was the flow rate used to infuse the nutrients and components to be tested for toxicity. This condition is very important in the biological environment. In microfluidic devices, shear stress is created by fluid flow injection due to several important aspects, such as channel dimensions and geometry, cell concentration, cell line type, and the way the flow rate is delivered, among others. Microfluidics provides a good way to mimic flows found in veins and small arteries, where the flow is usually unidirectional and laminar [82]. Shear stress can influence cell attachment [83], pathological response [84], and developmental biology [85].

The studies’ outcomes in toxicity screening using the CGG system in the microfluidics device showed comparable results to the conventional toxicity studies, and the efficiency evaluation techniques applied were mainly based on fluorescence signals, followed by spectrophotometry and brightfield microscopy, molecular methods, and other techniques (enzyme-linked immunosorbent assay—ELISA, 3-[4,5-dimethylthiazol-2-yl]-2,5 diphenyl tetrazolium bromide—MTT, and Western blot), showing the achievement of high efficiency in a faster way and the possibility of automation.

One of the limitations of this review was the lack of a detailed comparison of the complexity of the structures and geometries presented by the microfluidic devices developed in the studies. This analysis could help us better understand the significance of the micro-CGG on a global scale as well as the role that device design plays in the generation of the gradient and in each biological model that was investigated, but it was challenging due to the wide variation in the geometric arrangement and size of the studied biological models. Another limitation was the time frame used, 10 years, it was not sufficient to confirm whether there was a trend of CGG devices in relation to manufacturing characteristics, and previous gradient generation systems.

This systematic review also identified some methodological problems and research gaps, such as the relationship between the material used to make devices and the biological model or substance tested for toxicity, taking into account the benefits and drawbacks of each material, the sparse use of simulation procedures prior to device fabrication, and also the methodological care with regard to the duration of stable concentrations obtained by the CGG, which may compromise the accuracy of toxicity evaluation.

## 5. Conclusions

This systematic review showed a variety of toxicity assessment applications in the environmental and medical approaches through concentration gradient generation systems in microfluidic devices. Current studies have adopted new technologies and complex structures to customize the device according to the biological model, to achieve the best testing efficiency and to minimize typical microfluidics issues such as bubbles and shearing. The microfluidic gold-standard technique, soft lithography, using the polymer PDMS, was still the most frequently used, and the Christmas tree shape was the most prevalent CGG design, but alternative techniques and designs were employed to produce a larger variety of concentrations and drug combinations more precisely and more outcomes at once. Thus, the CGG microdevice is an alternative to common pipetting techniques for the evaluation of drugs’/substances’ toxicity in various biological organisms, bringing greater precision with a lower cost.

## Figures and Tables

**Figure 1 cells-11-03101-f001:**
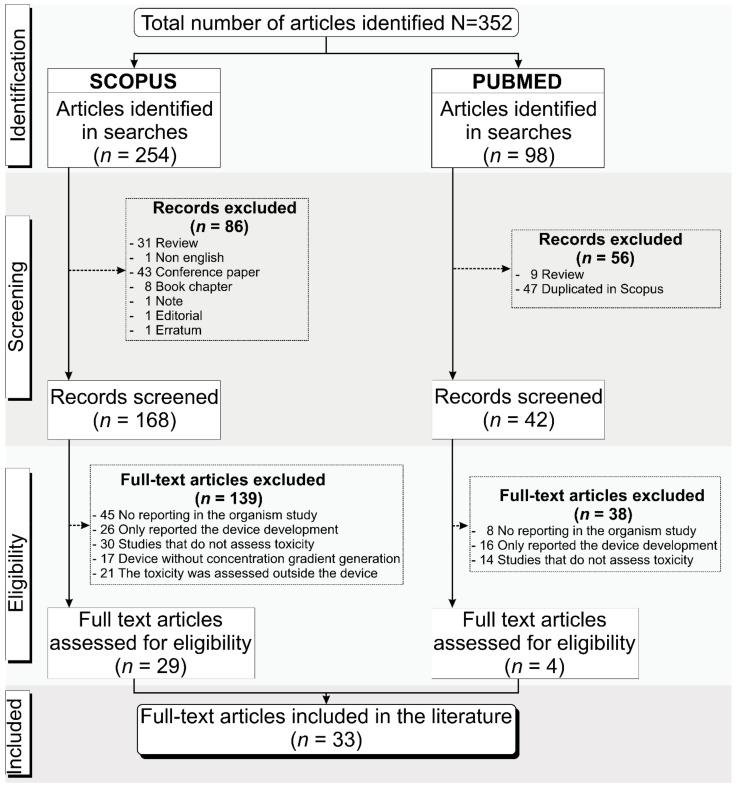
Schematic representation of the process for articles’ identification, screening, and eligibility for inclusion in this systematic review following the PRISMA guidelines.

**Figure 2 cells-11-03101-f002:**
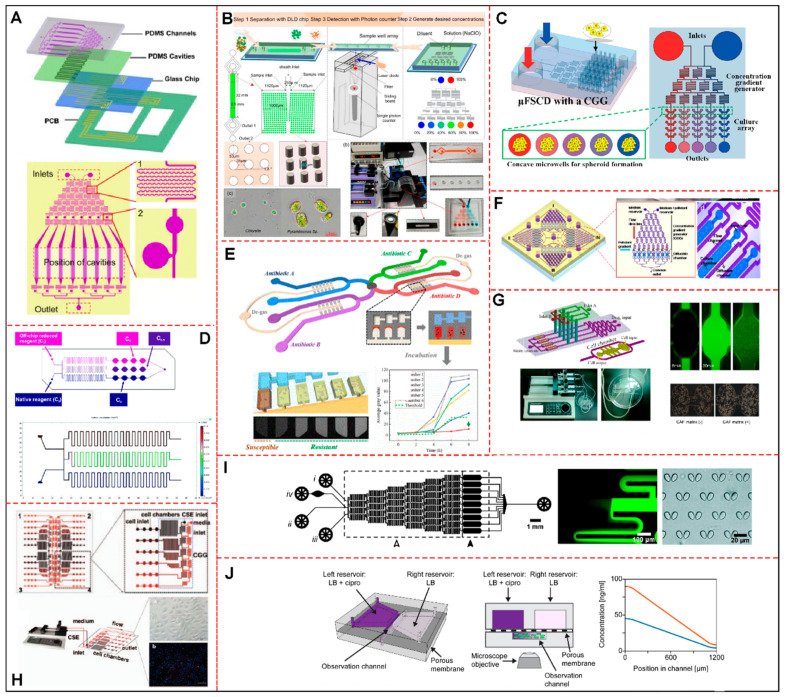
Schematic diagram of microfluidic devices with CGG system for toxicological analysis, representing some of the studies selected in this systematic review. (**A**) Representation of device layers, gradient generator structure, details of fluid mixing units and air bubble valves. Adapted with permission from [44], *Biosensors and Bioelectronics*. (**B**) Project showing a physical map of the CGG system and the photomicrograph of *Pyramimonas* sp. and *Chlorella*. Adapted with permission from [23], *Sensors* (Switzerland). (**C**) A schematic showing the design of a μFSCD with a concentration gradient generator. It exposed the structures, dimensions, and characteristics of the two layers, adapted with permission from [34], *Molecules*. (**D**) Construction of the Sphero Chip system proving the measurement principle of the experimental scheme and results of the computational modeling of a CGG structure. Adapted with permission from [41], *Lab on a Chip*. (**E**) The microdevice contains eight sets of C-Chambers, which can simultaneously enable eight sets of noninterfering ASTs with each other. Antibiotics can be preincorporated into the C chambers with a specific mass gradient. AST and MIC results can be obtained by comparing the fluorescence intensities between each set of C-Chambers. Adapted with permission from [16], *Biosensors and Bioelectronics*. (**F**) CGG microdevice used for toxicity tests based on marine phytoplankton motility containing four units connected to a central removable outlet. Shown is the enlarged image of the single-frame unit containing an upstream CGG and downstream diffusible cameras. Motility signals can be collected in real time. Adapted with permission from [25], *Marine Pollution Bulletin*. (**G**) Schematic design of the CGG microfluidic chip with cell chambers (top panel) and the chip manufactured with pumping machine (bottom panel). Chamber-diffused Rh-123 (green) and morphological characteristics of A549 cells with or without CAF matrix are shown. Adapted with permission from [37], *PLoS ONE*. (**H**) CGG containing four parallel operational modules including inputs CSE: 18 parallel cell chambers and 6 cell inputs. A CSE concentration gradient is shown from entry one to six, adapted with permission from [40], *Journal of Thoracic Oncology*. (**I**) Schematic overview of the microfluidic device with a CGG and chambers with passive hydrodynamic cell trap arrays. It shows details of branching and diffusional mixing of two fluorescent fluids with different concentrations and optical micrograph of cell traps in PDMS. Adapted with permission from [42], *Lab on a Chip*. (**J**) Schematic drawing of the CGG device, illustrating cross-section and theoretical profiles of Ciprofloxacin concentration in the observation channel. Antibiotic solutions with 3× MIC (blue curve) or 6× MIC (orange curve). Adapted with permission from [17], *Frontiers in Microbiology*. **Abbreviations:** CGG: concentration gradient generator; μFSCD: microfluidic spheroid culture device; ASTs: antibiotic susceptibility tests; Rh−123: Rhodamine; A549: adenocarcinoma human alveolar basal epithelial cell line; CAF: cancer-associated fibroblasts; CSE: cigarette smoke extracts; PDMS: polydimethylsiloxane; MIC: minimal inhibitory concentration.

**Figure 3 cells-11-03101-f003:**
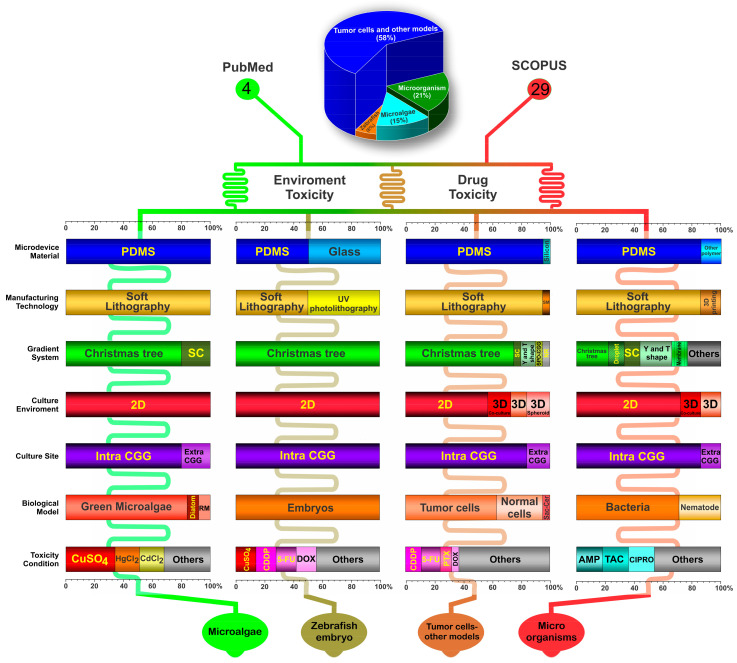
The systematic review identified 4 main types of organisms used for toxicity analysis using the CGG system in microfluidic devices: microalgae, zebrafish embryo, tumor cells and other models, and microorganisms. The figure shows the main important aspects (as percentages) regarding the microfluidic device material, manufacturing technology, gradient system, culture environment, culture site, biological model, and toxicity condition. Abbreviations: PDMS: polydimethylsiloxane; SC: serpentine channel; CGG: concentration gradient generator; RM: red microalgae; CDDP: Cisplatin; 5-FU: 5-Fluorouracil; DOX: Doxorubicin; SM: silicon; SPD-CGG: static-pressure-driven CGG; NR: not reported; Sac-Cer: Saccharomyces cerevisiae; PTX: Paclitaxel; AMP: Ampicillin; TAC: Tetracycline; CIPRO: Ciprofloxacin.

**Table 1 cells-11-03101-t001:** Characteristics, design, and fabrication of concentration gradient generator microfluidic devices for toxicity analyses.

Study	Year	Microfluidic Device				Mold		Device Assembly		Manufacturing	New Technologies
		Material	Layers	Technology	Geometry	Material	Fabrication	Cover	Sealing		
** *Microorganism* **											
Zeng, W. et al. [16]	2022	PDMS	2	Soft lithography	Chambers: 500 × 300 × 30 μm (L × W × H); Spacing between chamber: 40, 70, 100, 130, 160, and 190 μm	SU-8	UV photolithography	Glass	O_2_ plasma	In-house	Uses the properties of diffusion of gases based on vacuum pressure levels for drug gradient formation
Nagy, K. et al. [17]	2022	PDMS	3	Soft lithography	Upper layer: two trapezoid-shaped reservoirs (sides: 1.0, 0.5, 0.5, 0.65 cm (L), and 45 μL (Vol)); bottom layer: 0.04 × 1.2 × 10 mm (H × W × L), and 0.4 μL (Vol); overlapping area: 100 μm^2^	SU-8	UV photolithography	Glass	Partially cured PDMS	In-house	Use of a porous membrane for the diffusion of molecules from one reservoir to the other
Sweet, E. et al. [18]	2020	Polymer	NA	3D printing	Integrated vertical µ-mixers and channels between layers: 5000 × 7500 µm (L × D); bulbs: 1250 µm (D)	NA	NA	NA	NA	In-house	Unconventional 3D printing manufacturing with multidrug testing capability
Tang, M. et al. [19]	2018	PDMS	3	Soft lithography	Capillary valve: 2 × 0.2 × 0.3 mm (L × W × H); metering chambers: 10 μL (Vol), 2.5 × 4 × 1 mm (L × W × H)	PMMA	CNC machining	PDMS	Partially cured PDMS	In-house	Unconventional use of centrifugal microfluidics for the CGG; includes a laser photodetector and optical power meter
Zhang, B. et al. [20]	2014	PDMS	3	Soft lithography	Mixers: 200 × 60 µm (W × H); eight T-shaped channels: 280 × 60 µm (W × H); eight ellipses observing chambers: 3 × 4 × 0.5 mm (minor axis L × major axis L × H); inlet and outlet: 2 × 2 mm (D × H); ITO electrodes: 0.8 mm (W), 1350 A (H), 0.8 mm (spacing) Narrow path: 10 μm; PDMS on glass cover: 100 μm (T)	SU-8	UV photolithography	PDMS/ ITO glass	O_2_ plasma	In-house	Integration of an ITO glass layer for impedance system for worm-counting control
DiCicco, M. and Neethirajan S. [21]	2014	PDMS	NR	Soft lithography	Gradient mixing module: 20 × 100 × 18,750 μm (H × W × L) and the observation module: 20 × 1000 × 12,000 μm (H × W × L); bacteria inlet channel: 50 μm (W); inlet and outlet holes 750 μm (D)	SU-8	UV photolithography	Glass	NR	In-house	Vertical microchannel used for bacterial inoculum diffusion evaluation
Yang, J. et al. [22]	2013	PDMS	2	Soft lithography	Top layer: central reservoir (2.5 × 1 mm (D × H)) and eight inlets (2.0 × 1 mm (D × H)). Bottom layer: channels and culture chambers (2.0 × 0.5 mm (D × H)). Each channel is connected to a chamber by a thin “gate sill” (40 × 40 × 30 mm (W × H × T)	Glass plate/ copper plate	CNC machining	PDMS	O_2_ plasma	In-house	Worm dispenser system CGG microintegration
** *Microalgae* **											
Wang, Y. et al. [23]	2019	PDMS	1	Soft lithography	Specifications: 250 µm (sheath (center) inlet), 1125 µm (sample (side) inlet W), 1 mm (symmetrical micropost array W), 32 mm (overall L), 35 µm (vertical and horizontal spacing between the microposts), 50 µm (post D), 60 µm (channel H)	SU-8	UV photolithography	Glass	NR	In-house	Use of the principle of DLD for the separation of microalgae in the system
Han, B. et al. [24]	2019	PDMS	4	Soft lithography	Microchannels: 100 × 100 µm (deep × W); culturing chambers: 2 × 1.2 mm (L × W). The parallel channels and culturing chambers are 30 µm apart from each other and connected by diffusion channels (5 µm in depth)	SU-8	UV photolithography	PDMS	Plasma	In-house	Combinational–mixing–serial dilution design used to generate parallel gradients for mixing chemicals (binary/ternary/quaternary mixture) using an algorithm
Zheng, G. et al. [25]	2014	PDMS	1	Soft lithography	Reservoir: 5 mm (D); outlet holes: 1 mm (D); diffusible chamber connected between two parallel channels of each outlet of CGG: 500 µm (W), 2 mm (L). Channels and chambers: 50 µm (distance apart), 50 µm (H); chamber connected to flow channels by diffusion channels: 50 µm (W), 30 µm (distance apart), 2 µm (H)	SU-8	UV photolithography	PDMS	Air plasma	In-house	NR
Zheng, G. et al. [26]	2013	PDMS	3	Soft lithography	Three layers: the bottom flow layer containing flow channels to form an upstream CGG (100 × 50 μm (W × H)) and downstream parallel diffusion channels (1.2 × 2 × 0.1 mm (W × L × H)); polycarbonate membrane (10 μm (T), 1.2 μm (D pores)); the top culturing layer has structures of 16 isolated dead-end chambers for microalgae culture and imaging (l: (1.2 × 2 (W × L), hl: 100 μm (H)	SU-8	UV photolithography	PMMA	Partially cured PDMS/ O_2_ Plasma	In-house	NR
Zheng, G. et al. [27]	2012	PDMS	2	Soft lithography	Three cell culture chambers are connected between two parallel channels of each outlet of the CGG. The channels and chambers: 25 × 60 μm (distance apart × H), flow channels: 200 μm (W), chambers: 1.2 × 2 mm (W × L). The flow channels and chambers are connected by diffusion channels: 200 × 400 × 3 μm (W × L × H)	SU-8	UV photolithography	PDMS	Air plasma	In-house	3D culture system indirectly integrated by diffusion to the CGG
** *Tumor cells and other models* **											
Chennampally, P. et al. [28]	2021	Silicon	1	Silicon micromachining	Cell culture chamber: 1 × 1 × 0.1 mm (L × W × H); the cell culture chamber is covered with a thin glass cover slip ≈ 3 × 3 × 0.17 mm (L × W × H)	NA	NA	Glass	Anodic bonding; biomedical-grade RTV adhesive	In-house	Unconventional material that avoids chemical absorption and leaching problems introduced by PDMS
Yin, L. et al. [29]	2020	PDMS	OOC: 3 CGG: 2	Soft lithography	Kidney channel: 0.5 × 1 mm (W × H), cell culture: 14 mm (D); CGG- NR	PMMA	Laser cutting	Glass	O_2_ plasma	In-house	Direct interaction of an OOC with its own temperature control with a CGG
Jaberi, A. et al. [30]	2020	PDMS	2	Soft lithography	Microchambers for both devices: 1 mm (D); micropillar array: 80 µm (D of each pillar)	Etched silicon	UV direct writing photolithography	Glass	O_2_ plasma	In-house	NR
Zhao, X. et al. [31]	2019	PDMS	2	Soft lithography	Channel: 25 × 200 µm (H × W); culture chamber: 5850 µm (W)	SU-8	UV photolithography	Glass	O_2_ plasma	In-house	Use of hydrostatic pressure to conduct the fluid flow, with a pump-free microfluidic gradient generator
Qin, Y.X. et al. [32]	2018	PDMS	2	Soft lithography	Chip A: main channels and cell chambers 2 × 0.1 mm (D × H); Chip B: cell chambers 2 × 1 × 0.1 mm (L × W × H), central channel: 15 × 0.8 × 0.1 mm (L × W × H), traffic channels: 30 × 100 μm (W × H)	NR	NR	Glass	O_2_ plasma	In-house	Innovation in using two different integrated chips for CSE concentration generation
Luo, Y. et al. [33]	2018	PDMS	2	Soft lithography	NR	SU-8	UV photolithography	Glass	NR	In-house	NR
Lim, W. and S. Park [34]	2018	PDMS	2	Soft lithography	Chip dimensions: 4 × 3 × 0.8 cm (L × W × H); top layer: 6 mm (T), two inlets: 8 mm (D); gradient generator: microchannels 150 µm, culture array with fifty cell injection holes 700 µm (D) and five outlets 2 mm (D); bottom layer: 2 mm (T) with 50 concave microwells 400 × 200 µm (DxH)	SU-8	UV photolithography	PDMS	O_2_ plasma	In-house	Integration of µFSCD with a CGG
Jin, D. et al. [35]	2016	PDMS	3	Soft lithography	Inlet and outlet: 1.2 mm (D)	SU-8	UV photolithography	Glass	O_2_ plasma	In-house	Layer combination for the integration of spheroid cell culture and 2D culture with interaction over a porous membrane
Hong, B. et al. [36]	2016	PDMS and Paper plastic	3	Soft lithography	Inlets and outlets: 6 mm (D); 2 mm (channel W); chip dimensions: 49 × 47 mm (L × W)	SU-8	UV photolithography	Top and bottom plastic	Screw fastening	In-house	Paper-based chip
Ying, L. et al. [37]	2015	PDMS	3	Soft lithography	CGG: inlets 1.5 mm (D); microchannels 10 × 0.2 × 0.1 mm; cell culture chambers: 800 × 400 × 100 μm (L × W × H); inlet and outlet: 0.6 mm (D); upper PDMS layer: inlets 1.5 mm (D); perfusion channels: 200 × 100 μm (W × H)	SU-8	UV photolithography	Glass	O_2_ plasma	In-house	CGG fabrication utilizing vertical channel gravity for flow styling
Ju, S.M. et al. [38]	2015	PDMS	3	Soft lithography	Gradient generator and cell culture microchamber channel: 100 µm (H); pneumatic channel H: 150 µm	SU-8	UV photolithography	Glass	O_2_ plasma	In-house	Development of pump-free CGG with micropump system
Pasirayi, G. et al. [39]	2014	PDMS/PMMA	3	Soft lithography	Two PDMS layers 100 µm (T); control layer microchannels: 200 × 200 µm (W × H); CGG microchannels: 300 × 200 µm (W × H)	SU-8	UV photolithography	Glass; PDMS; PMMA	Clamping	In-house	NR
Li, E. et al. [40]	2014	PDMS	1	Soft lithography	CGG mixers and 6 main channels: 300 × 100 µm (W × H); 18 cell chambers: 2.5 × 0.1 mm (D × H)	SU-8 and positive photoresist	UV photolithography	Glass	O_2_ plasma	In-house	NR
Kwapiszewska, K. et al. [41]	2014	PDMS	1	Soft lithography	Microchamber: 50 μm (H); 18 microwells of 200 × 150 μm (D × H); microchannel: 200 μm (W)	PMMA; PDMS	CNC machining	PDMS	O_2_ plasma	In-house	Integration of spheroid culture with CGG compared to 2D culture
Fernandes, J.T.S. et al. [42]	2014	PDMS	1	Soft lithography	Serpentine-shaped microchannel: 50 × 9 μm (W × H)	SU-8	UV photolithography	Glass	PDMS	In-house	CGG array of hydrodynamic cell traps integration
Jastrzebska, E. et al. [43]	2013	PDMS	2	Soft lithography	Culture chamber: 1000 × 30 μm (D × H); microchannels: 100 × 50 μm (W × H)	Pro/Cap 50 and S1818	UV photolithography	Glass	O_2_ plasma	In-house	NR
Xu, Y. et al. [44]	2012	PDMS	4	Soft lithography	Cell culture and the cytotoxicity assay, PDMS cavities (2 × 8 × 2 mm (H × L × W)); eight groups of impedance sensing electrodes (20 × 30 μm (W × interelectrode distance)	SU-8;	UV photolithography	Glass	Thermal ball bonding; O_2_ plasma	In-house	CGG air bubble valves development to stop the fluid flow
Yang, C.G. et al. [45]	2011	PDMS	NR	Soft lithography	Six circular channel 150 × 40 μm (W × H); serpentine channels 80 × 40 μm (W × H); inlet 0.3 mm (D); wedge-shaped chambers for cell culture 7 × 100 × 360 × 40 μm (L × W (narrow side) × W (wider side) × H)	AZ P4620	UV photolithography	NA	Air plasma	In-house	NR
Jedrych, E. et al. [46]	2011	PDMS	2	Soft lithography	Matrix (5 × 5) of the culture microchambers 1000 × 30 μm (D × H) coupled with microchannels, creating the CGG; microchannels 100 × 50 μm (W × H)	Pro/Cap 50 and S1818	UV photolithography	Glass	O_2_ plasma	In-house	NR
** *Zebrafish embryos* **											
Li, Y. et al. [47]	2015	PDMS	2	Soft lithography	Circular channels: 200 × 50 µm (W × H); serpentine branch channels: 200 × 50 µm (W × H); solution inlets: 1 mm (D); cylinder-shaped chambers (7 mm (D), 2.5 µm (H))	Copper	CNC machining	Glass	O_2_ plasma	In-house	CGG can generate one blank solution, seven mixture concentrations, and eight single concentrations for each metal solution
Yang, F. et al. [48]	2011	Glass	2	UV direct writing photolithography	CGG on top slide: microchannels 120 × 30 μm (W × H), connective channels 300 × 30 μm (W × H), seven embryo inlets 1.3 mm (D); sandwiched culture chambers 4 × 1.7 mm (D × T of glass plate)	NA	NA	Glass	Anodic bonding	In-house	Reusable glass chip with natural hydrophilicity

Abbreviations: PDMS: polydimethylsiloxane; p82: polymer82; hw83: hydroxylated wax83; NA: not applicable; NR: not reported; OOC: organ-on-a-chip; CGG: concentration gradient generator; L: length; W: width; H: height; Vol: volume; D: diameter; T: thickness; SU-8: negative photoresist; S1813: positive photoresist; AZ P4620: positive photoresist; PMMA: poly(methyl methacrylate); CNC: computer numerical control; UV: ultraviolet; ITO glass: indium tin oxide glass; RTV: room-temperature vulcanizing; DLD: deterministic lateral displacement; µFSCD: microfluidic spheroid culture device.

**Table 2 cells-11-03101-t002:** Concentration gradient generator characteristics of microfluidic device.

Study	CGG Design			Concentrations Details					Flow Simulation	Stable Gradient Duration (min)	Advantages
	Method of Generation	Gradient System	Structure	Concentration Numbers	Concentration Type	Range	Validation	Stable Gradient Formation Time (sec)			
** *Microorganism* **											
Zeng, W. et al. [16]	Diffusion	Droplet generation	Eight C-Chamber sets, which can simultaneously enable eight AST sets without interfering with one another	8	Linear	KAN, AMP, TAC (1.2, 2.0, 3.5, 5.4, 7.3, and 13.1 μg/mL)	NR	40	NR	NR	Uses the properties of diffusion of gases by vacuum pressure levels for drug gradient formation
Nagy, K. et al. [17]	Diffusion	Membrane	Two trapezoid reservoirs in the upper layer and a rectangular reservoir in the bottom layer, with a porous membrane in between	2	Linear	CIPRO (3 × MIC = 48 ng/mL; 6 × MIC = 96 ng/mL)	NR	NR	COMSOL	NR	No shearing effect thanks to flow-free diffusion generation gradient
Sweet, E. et al. [18]	Convective	3D microchannel network	A tetrahedrally arranged network of nodal microchannel units, geometrically symmetric in 3D space and capable of generating three inherently symmetric fluid gradients	3	NR	TAC (0–0.5 mg/L); CIPRO (0–96 µg/L); AMK (0–16 mg/L) and buffer (control)	Rh	NR	COMSOL	NR	Integration of tetrahedrally arranged nodal combination–mixing–splitting units with a vertical u-mixing to obtain shearing-free and linear concentration flow
Tang, M. et al. [19]	NR	Centrifugal CGG	In two layers, at low spinning speed, with the help of centrifugal force, the fluidic content supplied by the source chamber will flow along the spiral channel and fill the metering chambers, while the redundant fluid will flow into the waste chamber	16	Linear	AMP (0–8 μg/mL; increases of 0.5 μg in each concentration)	Dye solutions	NR	NR	NR	Generates 16 accurate concentration levels, with slight variations, with the use of centrifugal force
Zhang, B. et al. [20]	Convective and diffusive	Serpentine channels and T-shaped channel	Eight T-shaped channels and eight ellipsoid observing chambers. Each T-shaped loading channel has been connected with a chamber and two separate inlets	8	Linear	0, 20, 50, 80, 100, 80, 50, e 20 µM (substance mixing)	FITC	NR	NR	NR	NR
DiCicco, M. and Neethirajan S. [21]	Convective and diffusive	Christmas tree and Y-junction	Microdevice consists of two solution inlets, a Christmas-tree-shaped CGG, one bacteria inlet, a straight main channel, and one outlet	9	Linear	NR	FITC	NR	NR	NR	NR
Yang, J. et al. [22]	Convective	Christmas tree	Radial worm dispenser with 32 trap-construction chambers and 4 multiple-gradient generators with a regular Christmas tree shape	8	Linear	0, 14.3, 28.6, 42.8, 57.1, 71.4, 85.7, and 100 mM (substance mixing)	FITC	NR	NR	NR	NR
** *Microalgae* **											
Wang, Y. et al. [23]	Convective	Christmas tree	Christmas tree CGG with two inlets and a rectangular cross-section	6	Linear	NaClO (250 ppm) (0, 50, 100, 150, 200, and 250 ppm); NaClO (500 ppm) (0, 100, 200, 300, 400, and 500 ppm)	NR	NR	NR	NR	Linear concentration was guaranteed thanks to the decrease in channel length at each level of the CGG structure
Han, B. et al. [24]	Diffusion	Snake model	The snake geometry is created by interactively folding a channel in an equal interval until the design specification is met according to the rules: L = (N + 1)ws + Ls − 3w, where w is the channel width, ws is the snake width, Ls is the snake length, N is the number of snake’s bends, and D is the snake density (Ls/N)	5	Linear	Cu (0.8, 1.6, 2.4, 3.2, 4 μM); Hg (0.8, 1.6, 2.4, 3.2, 4 μM); Zn (24, 48, 72, 96, 120 μM); Cd (16.5, 33, 49.5, 66, 82.5 μM)	NR	NR	CFD-ACE simulations	NR	Combination of linear channels with single-cell snake mixers to minimize design effort
Zheng, G. et al. [25]	Convective	Christmas tree	Four uniform structure units connected by a central outlet, each containing an upstream CGG with two inputs and downstream diffusible chambers	8	Linear	Hg (0, 0.43, 0.85, 1.28, 1.71, 2.13, 2.56, and 3.0 μM); Pb (0, 1.62, 3.24, 4.86, 6.48, 8.10, 9.72, and 11.34 μM); Cu (0, 0.625, 1.25, 1.875, 2.5, 3.125, 3.75, and 4.375 μM); Phenol (0, 1.29, 2.57, 3.86, 5.14, 6.43, 7.71, and 9.0 mmol/L); Phenol + Cu (0, 0.325, 0.65, 0.975, 1.3, 1.625, 1.95, and 2.275 mmol/L)	Rh	NR	NR	25	Prevents any active flow through the chambers and possible disruption of cell position, movement, or intercellular interaction
Zheng, G. et al. [26]	Convective	Christmas tree	Upstream serpentine channels and downstream parallel diffusion channels. Sixteen isolated dead-end chambers for microalgal culture and imaging	8	Linear	0, 1C/7, 2C/ 7, 3C/7, 4C/7, 5C/7, 6C/7 (each initial concentration—C)	Rh	320	NR	50	A torque-actuated valve system without use of an external power or pressure source
Zheng, G. et al. [27]	Convective	Christmas tree	Has a regular Christmas tree CGG shape that results in 8 gradients integrated with chemostatic chambers for microalgae culture	8	Linear	NR	Rh	180	NR	20	Use of different height than CGG flow channels of culture chambers to obtain no-return flow
** *Tumor cells and other models* **											
Chennampally, P. et al. [28]	Diffusion	Y-junction	The overall geometry is designed to mimic the primary aspects of the diffusion-based patterning of the neural tube	11	Linear	Rapamycin (0, 0.2, 0.4, 0.6, 0.8, 1.0, 1.2, 1.4, 1.6, 1.8, and 2.0 µM)	FITC	Established in <1800	COMSOL	Maintained indefinitely	Generates a gradient within the chamber that corresponds with SHH diffusion profiles to mimic neural tube
Yin, L. et al. [29]	Convective	Christmas tree	Christmas tree with two inlets	5	Linear	CDDP + CsA (0, 10, 20, 30, and 40 μmol/L); CDDP + Cim (0, 20, 40, 60, 80 μmol/L); GM (0, 10, 20, 30, and 40 mmol/L)	Dye solutions	NR	NR	NR	NR
Jaberi, A. et al. [30]	Convective	Christmas tree and micropillars	Microchambers (cell/drug) placed after each serpentine channel of the Christmas tree design. In another design, micropillars were also built into chambers to produce a gradient within the chambers	6	Linear	DOX (0, 6, 12, 18, 24, and 30 μg/mL)	DOX	NR	COMSOL	NR	Micropillars on each microculture chamber to produce a gradient within the chambers
Zhao, X. et al. [31]	Diffusion	Static-pressure-driven CGG	Consists of two rows, with seven inlets each, and eight mixing channels. The top row of inlets is connected to the last seven channels, while the bottom row is connected to the first seven channels	8	Linear	H_2_O_2_ (25 to 175 μM/500 μM (lethal dose))	FITC	NR	COMSOL	NR	Pump-free CGG generates a shear-free microenvironment with a tunable network to generate predefined biochemical gradients
Qin, Y. X. et al. [32]	NR	NR	Consists of 6 main channels and 18 cell chambers. The CGG module included five cascaded-mixing stages	6	NR	Theoretical proportion 0:1:3:5:7:9	NR	NR	NR	NR	NR
Luo, Y. et al. [33]	NR	Circular concentration gradient	Circular CGG with radial splitting–mixing–splitting–mixing processes	5	NR	0, 3C/4, C/2, 4C/3 and C (each initial concentration—C)	SF	NR	NR	NR	Radial splitting–mixing integration with a serpentine channel to obtain shearing-free and linear concentration
Lim, W. and S. Park [34]	Convective	Christmas tree	Christmas tree with two inlets and connected to a culture array	5	Linear	Irinotecan (0, 1.25, 2.5, 3.75 and 5 µM)	FITC	NR	NR	NR	NR
Jin, D. et al. [35]	Convective	Christmas tree	The top PDMS layer with two drug inlets integrated with six downstream 2D cell culture channels terminating at HUVEC inlets. The bottom PDMS layer has six 3D cell culture units	6	Linear	PTX (0.01–0.49 µg/mL); CDDP (0.09–4.95 µg/mL); 5-FU (2.3–390 µg/mL)	Rh	NR	NR	NR	NR
Hong, B. et al. [36]	Convective	Christmas tree	A regular Christmas tree shape within S-shaped mixers on two inlets and five outlets	5	Linear	DOX (4, 41, 90, 143, and 182.5 μg/mL)	Dye solutions	900	NR	NR	NR
Ying, L. et al. [37]	Convective and diffusive	Christmas tree and T-shaped channel	A combination of a linear CGG with two inlets and four downstream parallel cell culture units with two oval-shaped modules	4	Linear	PTX (0, 1.28, 2.59, and 4 μM)	Rh	1800	NR	NR	NR
Ju, S. M. et al. [38]	Convective	Christmas tree	Upstream CGG with six-step serpentine array to generate a diverse gradient at each step from two stock solutions	8	Linear	APAP (0, 5.7, 11.4, 17.1, 22.8, 28.5, 34.2, or 40 mM)	Dye solutions and FITC	NR	NR	NR	NR
Pasirayi, G. et al. [39]	Convective	Christmas tree	Two inputs connecting the regular Christmas tree shape with six outputs interconnected with four gradient culture microchambers, which have separate inlet and outlet reservoirs	6	Linear	PCN (0, 20, 40, 60, 80, and 100 mM); PTX + Aspirin (0, 2, 4, 6, 8, and 10 mM)	Dye solutions	NR	NR	NR	NR
Li, E. et al. [40]	Convective	Cascaded mixing	Consists of four parallel operating modules for simultaneous culture of four cell samples, and each module includes a CGG, 6 main channels, and 18 cell chambers	6	Linear	CSE (0, 2.37, 12.28, 19.86, 46.79, and 91.88%)	NR	NR	NR	NR	Cascaded-mixing stages that generated linear concentrations by adjusting the flow rate of two merging solutions in each stage through controlling channel length proportional to fluidic resistance
Kwapiszewska, K. et al. [41]	Convective	Serpentine channels	Spheroid culture microchambers were placed in an array of three serpentine channels, each containing four microculture chambers	3	Linear	5-FU (0, 0.125, 0.5, and 1 mM)	NR	Less than 20	COMSOL	NR	NR
Fernandes, J. T. S. et al. [42]	Convective	Christmas tree	Nine chamber sets, each containing hydrodynamic traps for yeast cells, and a chemical gradient generator has three inlets (solution inlets) that allow the insertion of chemical solutions of different compositions	9	Linear	Ascorbic acid (0, 0.13, 0.25, 0.38, 0.50, 0.63, 0.75, 0.88, and 1% (initial concentration percentage))	FITC	1	NR	NR	NR
Jastrzebska, E. et al. [43]	Convective	Christmas tree	A regular Christmas tree CGG shape that results in five gradients integrated on five meander modules each, totaling 25 culture microchambers	5	Linear	24 hrs: Celbx (39–83 μM) and 5-FU (93–202 μM); 48 hrs: Celbx (19–117 μM) and 5-FU (8–253 μM)	FITC	NR	NR	NR	Fully reusable; i.e., it can be used several times for various cell culture and cytotoxic experiments
Xu, Y. et al. [44]	Convective	Christmas tree	The device contained an upstream CGG with a regular serpentine mixer, eight air bubble valves, and downstream parallel cell culture chambers, aligned with the bottom cavities	8	Linear	NR	NR	NR	NR	NR	Fluid mixing units on CGG channels
Yang, C. G. et al. [45]	Convective	Christmas tree	Radial channel composed of multicircle channels and parallel branch channels. Latitudinal, six circular channels are arranged concentrically. Longitudinally, the serpentine branch channels are arranged symmetrically around each of the circular channels	65	Linear	5-FU (0–600 mg/mL); CDDP (0–400 mg/mL) and 5-FU + CDDP (0–600 mg/mL)	Rh	30	NR	NR	Integration of circular channels and serpentine branch channels to generate more concentration than conventional method
Jedrych, E. et al. [46]	Convective	Christmas tree	Consists of a matrix (5 × 5) of culture microchambers coupled with microchannels creating the CGG, which includes two inlets and five outlets	5	Linear	5-FU (0, 75, 150, 225, and 300 µM)	NR	NR	NR	NR	NR
** *Zebrafish embryos* **											
Li, Y. et al. [47]	Convective	Christmas tree	Composed of multicircle channels and parallel branch channels, latitudinally, three circular channels are arranged centrically and longitudinally, the serpentine branch channels; three inlets are located inside the first level, and a cylinder-shaped chamber array is located downstream of the branch channels in the outermost level	24 (8 gradients per drug)	Linear	NR	Eosin Y, FITC, and ethanol	NR	NR	NR	Centripetal geometry and the arrangement of concentric serpentine channels are able to generate mixing and single concentrations automatically
Yang, F. et al. [48]	Convective	Christmas tree	Contains a simple Christmas tree CGG with two inlets generating seven concentration gradients	7	Linear	Theoretical proportion 0, 16.7, 33.3, 50, 66.7, 88.3, 100 μg/mL	NR	96	NR	NR	NR

Abbreviations: NR: not reported; CGG: concentration gradient generator; AST: antibiotic susceptibility testing; HUVEC: human umbilical vein endothelial cell; PDMS: Polydimethylsiloxane; KAN: Kanamycin; AMP: Ampicillin; TAC: Tetracycline; CIPRO: Ciprofloxacin; MIC: minimal inhibitory concentration; AMK: Amikacin; µM: micromolar; mM: millimolar; ppm: parts per million; CDDP: Cisplatin; CsA: Cyclosporin A; Cim: Cimetidine; GM: Gentamycin; DOX: Doxorubicin; PTX: Paclitaxel; 5-FU: 5-Fluorouracil; APAP: Acetaminophen; PCN: Pyocyanin; CSE: cigarette smoke extracts; Celbx: Celecoxib; DI water: deionized water; FITC: fluorescein isothiocyanate; Rh: Rhodamine; SF: sodium fluorescein; PBS: phosphate-buffered saline; CDF-ACE: computational fluid dynamics; SHH: sonic hedgehog; ECIS: electric cell–substrate impedance sensing.

**Table 3 cells-11-03101-t003:** Biological model used for toxicity evaluation in CGG microfluidic device.

Study	Biological Model				Culture Environment						Toxicity Conditions		
	Origin	Type	Organism	Source	Culture Site	Biological Structure	Number of Organisms	Culture Medium	Temperature (°C)	Condition Environment	Stimulus/Drug ^(11)^	Flow rate; Diffusion Constant	Incubation Time (h)
** *Microorganism* **													
Zeng, W. et al. [16]	Bacterium	*E. coli k*12	*E. coli* 5α-GFP	NR	Intra-CGG	3D culture	5 × 10^6^ cfu/mL	LB broth medium	NR	N_2_ and O_2_	AMP (100 mg/mL; MIC: NR); KAN (10 mg/mL; MIC: 7.1 μg/mL); TAC (10 mg/mL; MIC: ~3 μg/mL) + DI water	NR; NR	8
			*E. coli* K-12					Alamar Blue (1 μg/mL) and LB broth medium (1:10)			MIC values: KAN 7.8 μg/mL; AMP 4.9 μg/mL; TAC 3.5 μg/mL		
Nagy, K. et al. [17]	Bacterium	*E. coli k*12 ^(1)^	W3110-GFP	NR	Intra-CGG	3D Co-culture (1:1)	10^5^ (10^3^ cell morphometric and localized)	LB broth medium and antibiotic-free LB	30	NR	CIPRO (3 and 6 x MIC)	NR; 6.87 × 10^–6^ cm^2^/s	48 and 72
			W3110-RFP										
Sweet, E. et al. [18]	Bacterium	*E. coli B*	BL21-DE3 ^(2)^	Agilent Technologies, CA, USA	Extra CGG	2D culture	5 × 10^5^ cfu/mL	LB broth medium	37	4% CO_2_	TAC, CIPRO, and AMK; buffer (control) combined each other	NR; NR	NR
Tang, M. et al. [19]	Bacterium	*E. coli B ^(2)^*	BL21-DE3	DBE-CEAS- Nanjing University, China	Intra-CGG	2D culture	10^6^ cfu/mL	LB medium + KAN (50 μg/mL)	37	NR	AMP	NR; NR	5
Zhang, B. et al. [20]	Nematode	*C. elegans*	CL2166 ^(3)^	NR	Intra-CGG	2D culture	1 worm/mL	NGM with OP50	Dark 20	NR	MnCl_2_ (100 mM) + vitamin E, resveratrol, or quercetin (100 μM), and K solution	5 μL/min; NR	48
			BZ555 ^(4)^										24
			N2 ^(5)^										
DiCicco, M. and Neethirajan S. [21]	Canine bacterium	*S. pseudintermedius*	MRSP A12	University of Guelph, Ontario Veterinary College, Canada	Intra-CGG	2D culture	~10^8^ cfu/mL	Columbia agar; TSB-G tubes at a 0.5 McFarland standard	35	NR	FO (16, 32, and 64 μg/mL)	10 μL/h; NR	4
Yang, J. et al. [22]	Nematode	*C. elegans*	Glp-4 (bn2ts)sek-1 (km4)	NR	Intra-CGG	2D culture	1–1.5 worms/μL	S. Aureus; 10% BHI –M9 medium; 5 μg/mL nalidixic acid	25	NR	AMX	10 μL/min; NR	for 12, 24, 36, 48, and 60
								E. coli op50; NGM + 5 μg/mL nalidixic acid					
			Glp-4 (bn2ts)sek-1 (km4)					S. Aureus; 10% BHI –M9 medium			AMX, aloe-emodin, rhein, and emodin with DMSO at 2%		48
** *Microalgae* **													
Wang, Y. et al. [23]	Green microalgae (Chlorophyta)	Marine microalgae	*Pyramimonas* sp.	LOFSRI, Dalian, China	Intra-CGG	2D culture	240 cells/μL	Enriched seawater medium	22–25	NR	NaClO (250 ppm)	6 μL/min; NR	12
			*Chlorella sp. (chl-1)*				580 cells/μL				NaClO (500 ppm)		
Han, B. et al. [24]	Green microalgae (Chlorophyta)	Marine microalgae	*P. subcordiformis (chl-6)*	KLMB, IOCAS, CAS, China	Intra-CGG	2D culture	>10^5^	F/2 medium	~25	60 μmol photon m^2^/s	CuSO_4_·5H_2_O; HgCl_2_, CdCl_2_·2.5H_2_O, ZnSO_4_·7H_2_O; single and binary mixing	1.5 µL/min; NR	1
Zheng, G. et al. [25]	Green microalgae (Chlorophyta)	Marine microalgae	*P. subcordiformis (chl-6)*	Chinese coast	Intra-CGG	2D culture	10^6^ individuals/mL	F/2 medium	25 ± 0.5	60 μmol photon/m^2^/s	CuSO_4_·5H_2_O (3 µmol/L); Pb(CH_3_COO)_2_.3H_2_O (11.34 µmol/L); HgCl_2_ (4.4 µmol/L) and phenol (9 mmol/L)	50 μL/min; NR	2
			*P. helgolandica var. tsingtaoensis (chl-5)*								CuSO_4_·5H_2_O (4.34 µmol/L); Pb(CH_3_COO)_2_.3H_2_O (13.3 µmol/L); HgCl_2_ (10 µmol/L) and phenol (12 mmol/L)		
Zheng, G. et al. [26]	Green microalgae (Chlorophyta)	Marine microalgae	*P. subcordiformis (chl-6)*	KLEMB, IOCAS, China	Extra CGG	2D culture	>10^5^	F/2 medium + CuSO_4_·5H_2_O	∼25	CO_2_/O_2_; 80 μmol photon/m^2^/s	CuSO_4_·5H_2_O (0–25 μmol/L)	1 μL/min; 6 × 10^−6^ cm^2^/s	72
			*P. helgolandica var. tsingtaoensis (chl-5)*								CuSO_4_·5H_2_O (0–40 μmol/L)		
			*Chlorella sp. (chl-1)*								CuSO_4_·5H_2_O (0–10 μmol/L)		
	Diatom (Bacillariophyta)		*Phaeodactylum tricornutum (bac-2)*								CuSO_4_·5H_2_O (0–23 μmol/L)		
	Red microalgae (Rhodophyta)		*Porphyridium cruentum (rho)*										
Zheng, G. et al. [27]	Green microalgae (Chlorophyta)	Marine microalgae	P. subcordiformis (chl-6)	NR	Intra-CGG	2D culture	10^6^ individuals/μL	F/2 medium + CuSO_4_·5H_2_O and CdCl_2_·2.5H_2_O	25	60 μmol photon/m^2^/s	CuSO_4_·5H_2_O (12.5 μmol/L); CdCl_2_·2.5H_2_O (225 μmol/L)	0.1 μL/min; 6 × 10^−6^ cm^2^/s	1.5
			P. helgolandica var. tsingtaoensis (chl-5)								CuSO_4_·5H_2_O (40 μmol/L); CdCl_2_·2.5H_2_O (500 μmol/L); single and Cu and phenol mixture		
** *Tumor cells and other models* **													
Chennampally, P. et al. [28]	Mice ^(6)^	Embryonic stem cell	ESC-WT	Primary cell	Intra-CGG	2D and 3D culture	10^6^–10^7^	Fresh medium, Geltrex	37	5% CO_2_	Rapamycin (1 µM)	~100 µL/hr; 4.9 × 10^−6^ cm^2^/s	168
			A315T										
Yin, L. et al. [29]	Human	Renal proximal tubule epithelial cells	RPTECs	Primary cell	Intra-CGG	3D co-culture	5 × 10^4^	High-glucose DMEM; ECM + collagen	37	5% CO_2_	CDDP, GM, CsA, and Cim	10–100 μL/min; NR	168
		Peritubular capillary endothelial cells	PCECs										
Jaberi, A. et al. [30]	Human	Epidermoid carcinoma	A431-DPNTP	Prof. Kathleen Green; NU; Prof. James K. Wahl, UNMC	Intra-CGG	3D co-culture	10^6^	GelMA; DMEM + FBS (10%) + P-S (1%)	NR	NR	DOX 98–102%	0.1 μL/min (bottom-top) and 0.2 μL/min (end to middle); NR	24
			A431-S2849GDP										
Zhao, X. et al. [31]	Mouse NIH/Swiss embryo	Fibroblast cell	NIH 3T3	NR	Intra-CGG	2D culture	3.4 × 10^5^	DMEM medium + FBS (10%)	37	5% CO_2_	Low and lethal dose of H_2_O_2_	0.2 nL/s; 4.9 × 10^−10^ m^2^ /s	120
Qin, Y. X. et al. [32]	Human	Bronchial epithelial cells	16HBE	SPF-EAC-DMU, China	Intra-CGG	2D culture	10^6^	RPMI-1640 serum free	NR	NR	CSE from two research-grade cigarettes	6 μL/min; NR	48
Luo, Y. et al. [33]	Rat	Insulinoma cell	INS-1	NICLR, CAM, China	Intra-CGG	3D culture	10^6^	RPMI-1640 + FBS (15%) + P-S (100 U/m) + BME matrix	37	5% CO_2_	Low (5.6 mmol/L) to high (25.5 mmol/L)—glucose plus glipizide	1.0 μL/min; NR	24, 36, 72, 96
Lim, W. and S. Park [34]	Human	Carcinoma colorectal	HCT116	ATCC	Extra CGG	Spheroid	2 × 10^4^	McCoy’s 5A Medium and Minimum Essential Media + FBS (10%) + P-S (100 U/mL)	37	5% CO_2_	Irinotecan (100 μM)	NR; NR	72
		Glioblastoma	U87-MG										
Jin, D. et al. [35]	Human	Endothelial cells	HUVEC	ATCC	Intra-CGG	2D culture	NR	DMEM/F12 medium + FBS (10%) + P-S (100 U/mL)	37	5% CO_2_	PTX, CDDP, and 5-FU single and mixture	NR; NR	24
	Human	Tumor cells	ACC-M ^(7)^	Dr. Wang (Guangzhou, China)	Extra CGG	Spheroid	2.5 × 10^7^	DMEM/F12 medium + BME matrix					
	Human		UM-SCC-6 cells ^(8)^	University of Michigan, USA	Extra CGG	Spheroid	2.5 × 10^7^	High-glucose DMEM + FBS (10%) + P-S (100 U/mL) + BME matrix					
Hong, B. et al. [36]	Human	Epithelial cervical carcinoma cells	HeLa	NR	Intra-CGG	3D culture	10^4^	DMEM + FBS (10%) + P-S (100 U/mL) + collagen type I	37	5% CO_2_	DOX (200 μg/mL)	NR; NR	2–8
Ying, L. et al. [37]	Human	Lung Tumor cell	A549	Cell Bank of Type Culture Collection of CAS, China	Intra-CGG	3D co-culture	10^6^	RPMI 1640 and IMDM + FBS (10%) + P-S (100 U/mL)+ BME matrix	37	5% CO_2_	PTX; PTX+ CAF; PTX + PI3K inhibitor; PTX + GRP78 inhibitor; PTX + CAF + PI3K inhibitor; PTX + CAF + GRP78 inhibitor;	10 mmHg/24 h; NR	24
		Fibroblast cell	HFL1										
Ju, S. M. et al. [38]	Human	Tumor liver cell	HepG2	Korean Cell line Bank, Korea	Intra-CGG	2D culture	2 × 10^6^	DMEM + FBS (10%) + P-S (100 U/mL) + fibronectin	37	5% CO_2_	APAP	1.7 μL/min; NR	24
Pasirayi, G. et al. [39]	Human	Breast tumor	MCF-7	Northern Institute for Cancer Research, Newcastle University	Intra-CGG	2D culture	2 × 10^5^	EMEM + Gln (2 mM/L) + nonessential amino acids (1%), FBS (10%) + P-S (100 U/mL) + A (1%) + fibronectin (100 µg/mL)	37	5% CO_2_	PCN (100 µM); PTX and aspirin	3.5–5 µL/min with 4 h intervals over a period of 24 h; NR	6
		Liver carcinoma cells	HepG2										
Li, E. et al. [40]	Human	Bronchial epithelial carcinoma cell	Primary	Patients of the First Affiliated Hospital of Dalian Medical University	Intra-CGG	2D culture	10^6^	Fresh medium	37	5% CO_2_	CSE	5–7 µL/min; NR	48
Kwapiszewska, K. et al. [41]	Human	Colon carcinoma cells	HT-29	ATCC	Intra-CGG	Spheroids	1 × 10^6^–5 × 10^6^	RPMI medium + FBS (5%) + L-Gln (1% of 25 mM) + S-P (1%)	37	5% CO_2_	5-FU	4.5 µL/min; change medium for 15 min	24
		Liver carcinoma cells	HepG2					EMEM medium + FBS (10%) + L-Gln (1% of 25 mM) + S-P (1%)					
Fernandes, J. T. S. et al. [42]	Saccharomyces cerevisiae	Yeast cells	VSY72 ^(9)^	NR	Intra-CGG	2D culture	~1.5 × 10^7^	SC + RAF (yeast nitrogen base without amino acids, 6.7 g/L); RAF (10 g/L); CSM without URA-TRP	30	NR	Ascorbic acid	0.3–0.5μL/min; NR	5
			Y4791 ^(10)^					SC–URA–TRP + GAL (1%) + FeCl_3_ (10 mM)			GAL (0 to 1%); RAF (1%); RAF (0.5%) + GAL (0.5%); GAL (1%)		5
											FeCl_3_ (0, 5, and 10 mM) + GAL (1%)		24
Jastrzebska, E. et al. [43]	Human	Lung carcinoma cell	A549 cell	ATCC	Intra-CGG	2D culture	1 × 10^6^	NR	37	5% CO_2_	Celbx (120 μM) and 5-FU (300 μM) single and mixture	15 μL/ min; change media 1.2 μL/min for 50 min	24 or 48
	Balb/c	Embryo cell	3T3 cells										
Xu, Y. et al. [44]	Human	Epithelial cervical carcinoma cells	HeLa	ATCC	Intra-CGG	2D culture	1.5 × 10^6^	DMEM + FBS (10%)	37	5% CO_2_	CDDP (0–20 μM)	4 μL/min; NR	24–48
		Colon carcinoma cells	RKO										
		Epidermoid carcinoma cells	CaSki					RPMI-1640 medium + FBS (10%)					
		HPV-related endocervical adenocarcinoma	SMMC-7721	PUMC, Beijing, China									
Yang, C. G. et al. [45]	Human	Uterine cervix cancer cell	HeLa	Key Lab of Cell Biology of Ministry of Public Health, PRC	Intra-CGG	2D culture	10^5^ cells/mL	DMEM + FBS (10%) + S-P (100 U/mL)	37	5% CO_2_	5-FU (600 mg/mL) and CP (400 mg/mL) single and mixture, and CDDP	2.0 μL/ min; NR	24–48
Jedrych, E. et al. [46]	Human	Lung carcinoma cell	A549	ATCC	Intra-CGG	2D culture	1 × 10^6^	RPMI 1640 medium + FBS (10%) + Glutamax (2 mM) + S-P (100 U/mL) + A (250 ng/mL)	37	5% CO_2_	5-FU	1.2 μL/min for 50 min; NR	24–48
		Colon adenocarcinoma cell	HT-29				3 × 10^6^						
** *Zebrafish embryos* **													
Li, Y. et al. [47]	Zebrafish	Zebrafish	Embryos	School of Life Sciences, SYSU, China	Intra-CGG	2D culture	10–12 eggs (3 hpf)	Ultrapure water medium + HNO_3_ (0.1 mol/L) + NaOH (0.1 mol/L)	28.5	O_2_	PbAc (1 mg/L); CuSO_4_ (0.1 mg/L)	10 μL/min–5 μL/min to 30 μL/min at each inlet; NR	48
Yang, F. et al. [48]	Zebrafish	Danio rerio	Embryos	School of Life Sciences, SYSU, China	Intra-CGG	2D culture	1 embryo/chamber	Embryo medium E3: NaCl (5 mM) + KCl (0.17 mM) + CaCl_2_ (0.40 mM) + MgSO_4_ (0.16 mM) per 100 mL distilled water	26 ± 1	Anoxia and normoxia	ADM (0–100 μg/mL)	4 μL/min; NR	1, 4, 12, 23, 24, 68, and 72 hpf
											DOX (0–100 μg/mL)		
											5-FU (0–100 μg/mL)		
											CDDP (0–100 μg/mL)		
											Vitamin C (0–100 μg/mL)		

Abbreviations: NIH: National Institute of Health; Balb/c: Bagg Albino Mouse; E. coli: Escherichia coli; C. elegans: Caenorhabditis elegans; S. pseudintermedius: Staphylococcus pseudintermedius; S. aureus: Staphylococcus aureus; HPV: human papillomavirus; GFP: Green Fluorescent Protein; RFP: Red Fluorescent Protein; BL21(DE3): Ampicillin-resistant Gram-negative E. coli; MRSP: Methicillin-resistant S. pseudintermedius; clh-: chlorophyll type; P. Subcordiformis: Platymonas Subcordiformis; P. helgolandica: Platymonas hel-golandica; ESC-WT: embryonic stem cell wild type; A315T: ESC mutant; RPTECs: renal proximal tubule epithelial cells; PCECs: peritubular capillary endothelial cells; A431-DPNTP: epidermoid carcinoma wild type; A431-S2849GDP: A431-targeted GFP-E-cadherin cells; NIH 3T3: NIH/Swiss mouse embryo fibroblast cell line; HBE: human bronchial epithelial cells; INS-1: rat insulinoma cell line; HCT116: colon cancer cell line; U87: glioma cell line; HUVEC: human umbilical vein endothelial cell; ACC-M: adenoid cystic carcinoma cell line; UM-SCC-6: human tongue squamous cell carcinoma cell line; HeLa: immortal cervical cancer cell line; A549: adenocarcinomic human alveolar basal epithelial cell line; HFL: human fetal lung fibroblast; HepG2: hepatocellular carcinoma cell line; MCF-7: Michigan Cancer Foundation 7—human breast metastatic adenocarcinoma cell line; HT-29: human colorectal adenocarcinoma cell line with epithelial morphology; RKO: poorly differentiated colon carcinoma cell line; CaSki: human papillomavirus type 16-positive cell line; SMMC-7721: hepatocellular carcinoma cell line; NR: not reported; AT: Agilent Technologies; CA: California; USA: United States of America; DBE-CEAS-NU: Department of Biomedical Engineering, College of Engineering and Applied Sciences, Nanjing University; ATCC: American Type Culture Collection; LOFSRI: Liaoning Ocean and Fisheries Science Research Institute; KLEMB: Key Laboratory of Experimental Marine Biology; IOCAS: Institute of Oceanology of CAS; CAS/CAM: Chinese Academy of Medical Sciences; NU: Northwestern University; UNMC: University of Nebraska Medical Center; SPF: specific-pathogen-free; EAC-DMU: Experimental Animal Center of Dalian Medical University; NICLR: National Infrastructure of Cell Line Resource; PUMC: Peking Union Medical College; PRC: China Medical University; SYSU: Sun Yat-sen University; CGG: concentration gradient generator; cfu: colony-forming unit; hpf: hours post-fertilization; LB: Luria–Bertani; KAN: Kanamycin; NGM: nematode growth medium; TSB-G: Trypic soy broth plus glucose; BHI: brain–heart infusion; F/2: general enriched seawater medium; DMEM: Dulbecco’s modified Eagle’s medium; ECM: extracellular matrix; GelMA: gelatin methacryloyl; FBS: fetal bovine serum; P-S: penicillin–streptomycin; RPMI-1640: Roswell Park Memorial Institute 1640 Medium; BME: basement membrane extractant; IMDM: Iscove’s modified Dulbecco’s media; EMEM: minimum essential medium Eagle; M9: M9 minimal medium; L-Gln: L-Glutamine P-S-A: penicillin–streptomycin–amphotericin B; SC: synthetic complete; RAF: raffinose liquid medium; CSM: complete supplement mixture; SC-URA-TRP: SC medium without uracil and tryptophan; GAL: galactose; AMP: Ampicillin; MIC: minimum inhibitory concentration; TAC: Tetracycline; DI water: deionized water; CIPRO: Ciprofloxacin; AMK: Amikacin; FO: Fosfomycin; AMX: Amoxicillin; DMSO: dimethyl sulfoxide; CDDP: Cisplatin; GM: Gentamycin; CsA: Cyclosporin A; Cim: Cimetedina; DOX: Doxorubicin; CSE: cigarette smoke extract; PTX: Paclitaxel; 5-FU: 5-Fluorouracil; CAF: cancer-associated fibroblasts; PI3K: Phosphoinositide 3-kinase; GRP78: Glucose-regulated protein 78; APAP: Acetaminophen; PCN: Pyocyanine; Celbx: Celecoxib; CP: Cyclo-phosphamide; ADM: Adriamycin. Note: (1) E. coli: strain JEK1036, comprising W3110-GFP: lacYZ:GFPmut2; W3110-RFP: (lacYZ:mRFP, known as JEK1037); (2) Recombinant E. coli BL21-DE3 (pET28a-GFP); (3) C. elegans: (dvIs19 [pAF15 (gst-4::GFP::NLS)]); (4) C. elegans: (egIs1 [dat-1p::GFP]); (5) Strain Seattle 1945: N2: wild type; (6) mutant C57BL/6J: B6.Cg-Tg(Prnp-TARDBP*A315T) 5Balo/J:B6.Cg-Tg(Hlxb9-GFP) 1Tmj/J; (7) ACC-M is the salivary gland adenoid cystic carcinoma (8) UM-SCC-6 is the human tongue squamous cell carcinoma cell line; (9) yeast strain VSY72 (can1-100 his3-11 15 leu2-3 112 GAL1pr-SNCA(WT)-GFP::TRP1 GAL1pr-SNCA(WT)-GFP::URA3 ade2-1); (10) yeast strain Y4791 (can1-100 his3-11 15 leu2-3 112 GAL1pr-SNCA(WT)-GFP::TRP1 GAL1pr-SNCA(WT)-GFP::URA3 ade2-1); (11) the stimulus/drug concentration refers to as the initial concentration for each substance.

**Table 4 cells-11-03101-t004:** The proposal, evaluation, and outcome of the CGG microfluidic device studies applied in toxicity screening.

Study	Study Proposal	Techniques for Evaluation	Outcomes	Microfluidics Advantages
** *Microorganism* **				
Zeng, W. et al. [16]	To perform an AST on a microfluidic device with lyophilized antibiotics	GFP fluorescence detection; Alamar Blue	The MIC values obtained in the device were consistent with the gold-standard BMD method tested in *E. coli* k-12: KAN was 7.8 μg/mL; TAC 3.5 μg/mL; and AMP 4.9 μg/mL, and *E. coli* 5α showed slightly lower levels	Simple, stable, and controllable operation, needing only simple equipment. The device can be stored for later use. Requires only small samples of the tested substance and very little incubation time. Provides high throughput for multiple AST assays at once
Nagy, K. et al. [17]	To study the emergence of resistant bacteria in spatial CIPRO gradients, then to perform the genomic sequencing to identify the key mutations that lead to antibiotic resistance	Fluorescence time-lapse microscopy; genomic sequencing and biofilm assay (96 wells)	Most genes affected in 48-h and 72-h were related to the bacterial envelope (rfaG, rfaE, rfaQ, and rfaC). There were similar mutations (in the marR and rfaG genes) and a 2–4× increase in MIC in cells, even without antibiotics and in antibiotic gradient for 48 h that can be explained by the environmental stress, and at 72 h the MIC was 1–30× higher	Microfluidics mimic the complexity of natural microenvironments for bacterial resistance research, facilitating the evolution of resistance and promoting genetic diversity, even before the antibiotics administration
Sweet, E. et al. [18]	To identify optimal drug compositions through MIC values of an AST for the treatment of antibiotic-resistant *E. coli* bacteria and 3D µ-CGG to allow a symmetrical gradient of fluids combined more than 2 drugs solution at time	Resazurin metabolic indicator and spectrophotometry (OD600)	The bacterial growth response and the drug MIC values were the following ~20% for TAC at ~0.26 mg/L, ~5% for CIPRO at ~50 µg/L, and ~30% for AMK at ~11 mg/L. Lower MIC values increased growth. With combined drugs, an antagonism effect between ~0.34 mg/L of TAC and ~28.8 µg/L of CIPRO occured, and a synergism effect with ~7.68 mg/L of AMK and ~48.8 µg/L of CIPRO was seen, and the value recommended was ~6.08 mg/L of AMK and ~65.3 µg/L of CIPRO in the infection treatment	Provides higher throughput when compared to traditional assays testing multiple antibiotics. The devices are customizable and can be rapidly and cheaply produced for immediate application in medical routine
Tang, M. et al. [19]	To generate discrete concentration levels through mixing predefined volumes of sample and diluent at different proportions automatically	Absorbance by spectrophotometry	The number of *E. coli* bacteria did not increase after 3 h of exposure to ≥3.5 µg/mL of the MIC value of AMP	MIC produces results much more rapidly than traditional methods automatically, saving labor time. When integrated with optical detection units, it is more compact and cheaper than commercial spectrometer-based systems. Able to perform multiple processes simultaneously and has a fully customizable concentration gradient
Zhang, B. et al. [20]	To encapsulate a number of worms into the individual chamber and investigate the diverse behavioral responses to manganese toxicity	Fluorescence images by stereomicroscopy	The worm’s motility impairment was dose- and time-dependent when exposed to manganese; high concentrations can cause effects of DAergic neurodegeneration and cell death in the worms, and the natural antioxidants can protect against manganese-induced toxicity	Semi-automized processes. The microfluidic chamber design permits the formation of restricted habitats for the organisms, the administration of precise chemical stimuli, and their reaction assessment by conventional microscopy due to the optical transparency of the device’s materials. Additionally, it has a low cost, good biocompatibility, and versatility of chip
DiCicco, M. and Neethirajan S. [21]	To evaluate the in vitro activity of fosfomycin against MRSP biofilms, to determine the MBEC	SYTO 9 dye from a LIVE/DEAD^®^ BacLightTM bacterial viability kit	The MBEC value was 8.6 ± 2.1 μg/mL of FO, and the concentration of FO needed to remediate biofilm-embedded cells of MRSP A12 is 8.1 ± 0.9 μg/mL	Facilitates fast analysis of bacterial resistance, pointing to the correct therapeutic conduct
Yang, J. et al. [22]	To perform an in vivo antimicrobial screening assay and investigate antibacterial activity of some compounds of rhubarb (aloe-emodin, rhein, and emodin)	Plasmalemma fluorescent probe DiI by stereomicroscope. The lifespan is tested by LT_50_	The worm’s LT_50_ in M9 buffer, in 20% and 10% of BHI-M9 medium, was 60, 24, and 36 h. Exposure to *S. aureus* for 36 h exhibited suitable virulence to kill worms. At ≤36 h, some infected worms died due to their intestinal lumen filling with a large number of *S. aureus*, being killed in 5 days. At 48 h, the optimum AMX treatment time, the infected animals were rescued to varying degrees and treated with different concentrations of AMX (0–100 mg/mL); this was carried out in a dose-dependent manner and increased worm survival by at least 1.5-fold with an MIC of 4.0 mg/mL. Rhubarb inhibited the growth of *S. aureus*, and their MIC values were 7.5, 16.0, and 6.3 mg/mL, respectively, rescuing infected nematodes 1.0–2.0 fold more often at low concentrations, and killing them in higher concentrations (0.60 mg/mL)	Automized assay. Simultaneous generation of multiple concentrations. Reduces manual labor, reagent consumption, and time of analysis. Simultaneous assessment of antibiotic activity and toxicity of these drugs to the host, in vivo
** *Microalgae* **				
Wang, Y. et al. [23]	To perform DLD separation associated with the possibility of generating different desired concentrations of NaClO solution, using a single integrated photon counter	Chlorophyll fluorescence	*Pyramimonas* sp. viability decreases rapidly in the first 8 min, 8% after 20 min at 250 ppm of NaClO, and to almost zero at 20 min at 280 ppm. The *Chlorella* required a 500 ppm NaClO for complete inactivation within 20 min	Rapidly generates accurate concentrations. The device, compared to traditional methods, is more compact, cheaper, and more efficient, allowing the assay to be automized, and does not pollute
Han, B. et al. [24]	To assess metals’ toxicity to microalgae (copper, mercury, zinc, and cadmium) alone or in a binary/ternary/quaternary mixture	Brightfield microscope	*P. subcordiformis* motility inhibition increased with exposure to the increasing concentration of single pollutants of Cu, Hg, Zn, and Cd for 1 h. Hg was the most toxic, followed by Cu and Cd, and Zn was the least toxic. After 1 h, the metal mixture of Hg, Cu, and Cd with Zn was more damaging than Cu with Zn, Cd, or Hg	Offers higher-throughput alternative to conventional methods and might be employed for other types of assays
Zheng, G. et al. [25]	To assess the marine phytoplankton motility and investigate the pollutants’ toxicity effect (Hg, Pb, Cu, and phenol)	Movement tracking by CASA system: MOT, VCL, VAP, and VSL	After 2 h, the MOT data of Hg, Pb, Cu, and phenol showed them to be 2, 1.5, 2, and 1.2 times more toxic independently. The Cu and phenol mixture inhibited MOT and VSL in the range from 0 to 2.275 toxic units, being dose-dependent mainly for *P. subcordiformis* and *P. helgolandica var. tsingtoaensis*	Incorporation of multiple technologies in one assay. Offers high throughput, automation, low sample consumption, and shorter times. Automation of image acquisition
Zheng, G. et al. [26]	To assess multibiological model in the Cu toxicity test by measurements of cell division rate and esterase activity	Cell viability by cell autofluorescence and esterase activity by FDA	*P. subcordiformis* had the best condition for chemostatic culture (max 15 days). The microalgae growth decrease was dose-dependent on Cu concentration, *Chlorella* was more sensitive to Cu (EC_50_ of 5.52 μmol/L), and *P. helgolandia var. tsingtaoensis* was more resistant to Cu (EC_50_ of 20 μmol/L)	Simplifies toxicity assays. The device allows for easy customization of culturing conditions. Can also be rapidly fabricated
Zheng, G. et al. [27]	To assess the chemostat-based cell immobilization through metals’ (Cu and Cd) toxicity and motility	Bright-field microscope	The microalga motility inhibition was dose-dependent on Cu and Cd; *P. helgolandica var. tsingtaoensis* was more resistant than *P. subcordiformis*, for completed motility inhibition (28.60 versus 8.95 μmol/L of Cu and 357.15 versus 196.45 of Cd) using %MOT, VCL, VAP, and VSL data. Cu had a more toxic effect than Cd	Simplifies and accelerates toxicity assays
** *Tumor cells and other models* **				
Chennampally, P. et al. [28]	To evaluate the effectiveness of rapamycin in rescuing the MN of ALS	Immunostaining for GFP and TDP-43; Western blot	ALS-affected motor neuron survival can be increased by 40.44% in a rapamycin dosage range between 0.4 and 1.0 µM	As it is compatible with traditional techniques, they can be combined to obtain the advantages of both. Enhances throughput and results in the entire assay on only one cell culture. Enables multiple simultaneous tests, and has the capacity to stimulate cells to adopt spatial distribution and morphology similar to those in vivo
Yin, L. et al. [29]	To predict the nephrotoxicity induced by CDDP, GM, and CsA in renal chip	Calcein-AM/PI and CCK-8 assay	Cell viability was higher in static than fluidic co-culture condition. The cell viability was dose-dependent for all drugs. Cim neutralized and reduced the toxicity of CDDP, thus improving the survival rate of renal cells	Automation of multiple processes. Studies can be performed on models which reproduce key features of an organ’s physiology. Microfluidic devices can bring standardization, automation, and a reduction in costs to drug assays. They can also accelerate the whole process and lessen the impact of human bias
Jaberi, A. et al. [30]	To assess the mechanical and chemical stresses in skin cancer cell DOX	Live/dead (Calcein AM/ ethidiumhomodimer 1)	Cells showed a well-distributed morphology in the chambers and high viability (95%) without fluid flow. The effect of shear stress slightly reduced cell viability (88%) and also led to an increase in DOX concentration	Microfluidic devices may offer better conditions for 3D cell culturing and co-culturing. A single, versatile, device suitable for the evaluation of different conditions, while guaranteeing high throughput
Zhao, X. et al. [31]	To generate a shear-free microenvironment for long-term cell culture and adaptive cytoprotection analysis with a pumpless hydrogen peroxide gradient generator	Apoptosis by Annexin-V-FITC and PI	More stable and precise biochemical gradient by static pressure. Pretreatment of low-dose H2O2 protected NIH 3T3 cells against cytotoxicity. An H2O2 lethal dose results in 27.72% of apoptosis. Pretreatment for 24 h with lethal hydrogen peroxide exposure arrested the apoptosis in a dose-dependent manner. Apoptosis ratio decreased to ~27, ~22, and~14% with 25, 75, and 175 μM, respectively	Simple operation, without the need for external equipment and easy fabrication. A portable device which provides stable concentration gradients and is suitable for long-term cell culture, due to its low shearing effect
Qin, Y. X. et al. [32]	To detect the role of the HHS in CSE-induced malignant transformation of 16HBE	Apoptosis by fluorescence (Hoechst 33342), Western blot	16HBE CSE-induced cell apoptosis was dose-dependent, high doses (≥19.86%) promoted cell apoptosis, low doses (≤12.28%) promoted less apoptosis and continued cell growth (>80% cell viability). The best concentration for CSE stimulation was 12.25%, and after 15 weeks, some cells displayed condensed nuclei and abnormal nuclear-to-cytoplasmic ratios, atypical mitoses, and later a loss of contact inhibition. These alterations were not apparent in the cells treated with cyclosporine	Provided greater efficiency, accuracy, lower time, high throughput, and constant control of microenvironmental conditions via computer programs (automation), simple operation, and low costs of construction compared to traditional methods. Emulates the in vivo cell microenvironment and permits the dynamic observation of their growth
Luo, Y. et al. [33]	To drug screen for diabetes with glipizide in 3D INS-1 high-glucose cell model through the circular CGG	MTT, Calcein-AM/PI, Ultrasensitive Rat Insulin ELISA Assay kit	After 24 h addition of glipizide, the decrease rate of inhibition rate with glipizide concentration was 0.5916 and 0.3183 for 3D and 2D models, respectively, and after 48 h, it was 0.9133 and 0.4817 for 3D and 2D models, respectively. The 3D model was more sensitive than the 2D model and produced a greater insulin production response in diabetes drug screening	High throughput. The use of the 3D cell model, facilitated by the device, was shown to produce better results than the traditional method. Multiple parallel assays can be conducted
Lim, W. and S. Park [34]	To develop a µFSCD with a CGG that enables cells to form spheroids and grow in the presence of cancer drug gradients	Live/dead	The HCT116 cells’ viabilities are drug dose dependent, their viability decrease (63%) after 5 days of 5 μM irinotecan treatment (highest concentration), while the cell viability in the control was 98%.	The device facilitates homogenous spheroid generation. Allows for high-throughput and multiple parallel assays. Its CGG system makes the generated concentrations easily calculable. The materials used allow the gradients formed to be maintained for long periods and the observation to be made using a conventional optical microscope
Jin, D. et al. [35]	To assess drug sensitivity in spheroid head and neck perivascular tumor model and toxicity in endothelium	Hoechst 33342, PI, and immunostaining	The IC_50_ values of PTX, CDDP, and 5-FU for 3D-UM-SCC6 were 0.54, 5.5, and 454 μg/mL, respectively, and for ACC-M, they were 0.45, 5.2, and 400 μg/mL, respectively, being higher than in 2D culture. Low concentrations of PTX or 5-FU combined with CDDP had similar effects to high concentrations of a single drug on tumor cells and low cytotoxicity to HUVEC, leading to ~50% apoptosis of tumor cells, and already high concentrations of combinations were toxic to HUVEC cells. Different patients’ tumor cells showed relatively high sensitivities to both combinations with ~ 60% survival, while others showed low sensitivity with 80% cell survival	Allows for in vivo administration of drugs to be emulated. Microfluidic devices are better suited for the culture of spheroids, providing better results than conventional 2D culture methods. High throughput, lower costs, maintenance of concentration gradients for long periods of time, and real time analysis are features provided by the microdevice. If needed, more than one drug could be loaded into the device for testing
Hong, B. et al. [36]	To drug screen with CGG on a paper-based device	Live/dead (Calcein AM/PI; Prestoblue)	After 8 h, the cell viability was >50% with 50 µg/mL DOX and 20% with 200 μg/mL DOX	Allows for multiple simultaneous assays under different drug concentrations to be conducted, as well as automation and a reduction in costs and reagent volumes, increasing the overall efficiency
Ying, L. et al. [37]	To assess the impact of CAF or HGF on the Met/PI3K/AKT phosphorylation, GRP78 expression and PTX-induced apoptosis in A549 cells cultured in the 3D matrix	Viability (Rhodamine-123); immunofluorescence; Western blot; immunohistochemistry; apoptosis assay (Hochest33342 e PI); ELISA	HGF in the CAF matrix activated the Met/PI3K/AKT and up-regulated GRP78 expression, promoting chemoresistance to PTX-mediated apoptosis in A549 cells. PI3K and GRP78 inhibitors elevated PTX action in cell viability: 90%, 95%, and 100% at 1.28, 2.59, and 4 μM PTX, respectively	High throughput, high sensitivity, reduced substance volumes and overall experiment time. Emulates natural cell microenvironments
Ju, S. M. et al. [38]	To investigate APAP cytotoxicity through linear/diffusive-mixing-based CGG on HepG2 cells	Live/dead (Calcein AM/ ethidium homodimer 1)	The device showed more sensitivity in toxicity tests than in the 96-well culture (IC_50_ of 17.8 versus 22.8 mM, respectively), being 128% higher and >1800% less time-consuming due to the use of an automated LabVIEW system that refreshes APAP on the target cells every 4 h	Compared to the 96-well culture system, cells showed higher sensitivity to the substance tested, leading to the conclusion that the microdevice produces more accurate results. Time spent, as well as reagent and sample consumption, are reduced. Provides high throughput, integration of several techniques in one assay, and automation
Pasirayi, G. et al. [39]	To chemotherapeutically screen for PCN, PTX, and aspirin singly and combined in two types of tumor cells	Calcein AM	Concentrations of PCN and PTX LC_50_ on MCF-7 were ~60 and 0.63 μM, higher than in traditional culture (~51 and 0.55 μM), respectively. HepG2 showed the same results with high resistence to PCN (100 μM) with 70% of viability. A total of 0.2 µM of PTX reduced cell viability to 83%, while 4 mM aspirin alone reduced cell viability to 84%. PTX plus aspirina had a higher effect on the loss in cell viability than PTX alone	Cells cultured in the microdevice showed more growth after exposure to drugs, compared to those cultured in 96-well culture plates. Has a low cost and provides the possibility of testing drug combinations
Li, E. et al. [40]	To investigate the potential mechanisms underlying tumor-like transformation of continual exposure of primarily cultured human bronchial epithelial cells to CSE	Hoechst33342 and propidium iodide (PI); ROS Assay kit; immunofluorescent assay (GRP78, NF-κB, and PI3K) (E-cadherin and Vimentin); Western blot	Lower doses (2.37–12.28%) of CSE stimulated cell proliferation, but not cell apoptosis, and higher doses (19.86–91.88%) induced cell apoptosis. All analyses were one-way and dose-dependent, as well as the results for ROS	Emulates heavy smoking in humans and the lung microenvironment, making the device ideal for experiments of this kind due to its dimensions, its material properties, and the steady flow of the medium, therefore generating more accurate results. Additionally, it allows for parallel assays with diverse conditions, minimizing possible errors
Kwapiszewska, K. et al. [41]	To screen for anticancer drug and chemoresistance phenomena using the SpheroChip and assessing metabolic activity via dynamic changes in two types of tumor cells	Live/dead (Calcein AM/PI); Fluorescent resorufin (metabolic activity)	The growth of HepG2 spheroids was slightly higher than that of HT-29 inside a chip. HT-29 spheroid had normal metabolic activity until 20% 5-FU (0.125), being resistant to higher concentrations of 5-FU (up to 1 mM) compared to Petri dish culture, and exhibited a strong decrease in metabolic activity of 49% compared to the control (at 24 h)	Provides controllable conditions for 3D culture and the monitoring of the effects of the substances tested for long periods of time, which allows for time-dependent analysis, unlike conventional methods. The device’s fabrication and operation are simple, and it reduces costs and time of experiments
Fernandes, J. T. S. et al. [42]	To study aSyn production and aggregation in Saccharomyces cerevisiae using an elastomeric microfluidic device exposed to iron and ascorbic acid	Live-cell imaging; tracking the behavior of single cells by fluorescence image and α-synuclein (aSyn) production	The proportion of single cells trapped was higher for more loosely packed traps (43% for x = y = 20 μm). FeCl_3_ induced the formation of aSyn inclusions in a concentration-dependent manner, and ascorbic acid reduced the formation of aSyn inclusions in Y4791 yeast cells	The device enables the creation of controllable microenvironments with precise conditions and, also, the use of a minute quantity of solutions for creation of the concentration gradient, as well as the tracing, over time, of individual cells’ responses, unlike traditional methods. Compared to manual mixing of solutions, the CGG is prone to less mistakes, and is faster and less complicated
Jastrzebska, E. et al. [43]	To assess drug combinations of He and 5-FU anticancer on normal mouse embryo cells (Balb/c 3T3) and human lung carcinoma cells (A549)	Live/dead—Calcein AM/PI	Celbx and NSAID inhibited the growth of cancer cells and indicate anticancer properties. After cells’ incubation with Celbx, the viability of A549 cells was lower than normal Balb/c 3T3 cells, and Celbx plus 5-FU enhanced antitumor activity	The CGG made it possible to obtain multiple combinations of the tested substances automatically and simultaneously, while also improving the repeatability
Xu, Y. et al. [44]	To assess on four tumor cell lines (HeLa, CaSki, RKO, and SMMC-7721) and the cytotoxicity of the anticancer drug CDDP	Impedance sensing, fluorescent dye (FICT/PI)	EC_50_ of CDDP for CaSki and SMMC-7721 cells was below 4 μM and above 16 μM for HeLa and RKO cells. So, CaSki and SMMC-7721 cells showed more severe toxic responses to CDDP treatment compared to the other two cell lines	The device reduces reagent and sample consumption, cost and time of experiment, and enables automation, while providing high-throughput, label-free, and dynamic detection of the effects of substances tested
Yang, C. G. et al. [45]	To assess HeLa apoptosis of the single and combined effects of two drugs through combinatorial, quantitative, and predictable concentration gradient by repeated splitting and mixing	DAPI, annexin V-FITC/PI apoptosis detection kit	Cellular morphological changes with the increase in drug concentration: cell shrinkage, increase in cell granularity and chromatin condensation, and the most apoptosis characteristics. The apoptosis effect induced by CDDP was more obvious with the increase in stimulation time and concentration	The CGG produces an extremely wide range of stable, customizable, and repeatable concentrations. It also possesses a compact design and provides high throughput, while reducing time of analysis
Jedrych, E. et al. [46]	To assess the 5-FU cytotoxicity on two human cancer cell lines	Calcein AM/PI	After 24 h, cell death by 5-FU increased in a concentration/time-dependent manner, inhibiting the survival of both cell types; HT-29 cells were less sensitive than A549 cells. The strongest inhibition, approaching 80% after 48 h of incubation, was observed for A549 cells exposed to 300 μM 5-FU	The device allows for the execution of different methods of toxicological evaluation, as well as automation of processes. It also permits the simultaneous cultivation of cells with different characteristics and the lowering of costs and time needed
** *Zebrafish embryos* **				
Li, Y. et al. [47]	To perform metal safety evaluations and poison screening using embryos as vertebrate models	Morphological and behavior analyses; body length measured	Pb and Cu revealed an effect at 22 hpf, mortality at 24 hpf, heart rate and body length at 96 hpf, being concentration-dependent. The teratogenicity of Pb and Cu in zebrafish embryos and mixed metals induced more severe toxicity with several types of malformations	NR
Yang, F. et al. [48]	To describe a phenotype-based whole-organism model to assess the developmental toxicity and teratogenicity of anticancer drug-induced zebrafish embryos	Stereomicroscopy	ADM and CDDP had similar toxicity and teratogenicity in 4 hpf embryos, and 5-FU was halved under the same conditions. These effects vary according to developmental embryo stages, mainly for DOX, which exhibited obvious time/dose-dependent toxicity and LD 50 = 91.7 μg/mL. The embryos treated with vitamin C were not damaged	Allows for high throughput, combination of technologies, and automation

Abbreviations: AST: antibiotic susceptibility testing; CIPRO: Ciprofloxacin; MIC: minimal inhibitory concentration; µ-CGG: concentration gradient generator microdevice; MRSP: Methicillin-resistant *Staphylococcus pseudintermedius*; MBEC: minimum biofilm eradication concentration; DLD: deterministic lateral displacement; MNs: motor neurons; ALS: amyotrophic lateral sclerosis phenotype; CDDP: Cisplatin; GM: Gentamycin; CsA: Cyclosporin A; DOX: Doxorubicin; HHS: hedgehog signaling system; CSE: cigarette smoke extract; 16HBE: human bronchial epithelial cells; INS: insulinoma cell line; µFSCD: microfluidic spheroid culture device; CAF: cancer-associated fibroblasts; HGF: hepatocyte growth factor; P13K/AKT: Phosphoinositide 3-kinase; GRP78: Glucose-regulated protein 78; PTX: Paclitaxel; A549: adenocarcinomic human alveolar basal epithelial cell line; APAP: Acetaminophen; HepG2: hepatocellular carcinoma cell line; PCN: Pyocyanin; Celbx: Celecoxib; 5-FU: 5-Fluorouracil; Balb/c: Baag Albino Mouse; HeLa: immortal cervical cancer cell line; CaSki: human papillomavirus type 16-positive cell line; RKO: poorly differentiated colon carcinoma cell line; SMMC-7721: hepatocellular carcinoma cell line; GFP: Green fluorescent protein; OD600: optical density at a wavelength of 600 nm; SYTO9: fluorescent nucleic acid stain; Dil: 1,19-dioctadecyl-3,3,39,39-tetramethylindocarbocyanine per-chlorate; LT_50_: lethal time for 50% of a population; CASA: computer-assisted sperm analysis; MOT: motile percentage; VCL: curvilinear velocity; VAP: average path velocity; VSL: straight-line velocity; FDA: fluorescein diacetate; TDP-43: transactive response DNA-binding protein 43 kDa; Calcein-AM/PI: BioReagent, suitable for fluorescence; CKK-8: cell counting kit-8; Annexin-V-FITC: apoptosis detection kit; PI: propidium iodide; MTT: 3-[4,5-dimethylthiazol-2-yl]-2,5 diphenyl tetrazolium bromide; ELISA: enzyme-linked immunosorbent assay; ROS: reactive oxygen species; NF-KB: nuclear factor kappa-light-chain-enhancer of activated B cells; FICT: fluorescein isothiocyanate; DAPI: 4′,6-diamidino-2-phenylindole; BMD: broth microdilution; *E. coli: Escherichia coli;* KAN: Kanamycin; TAC: Tetracycline; AMP: Ampicillin; AMK: Amikacin; DAergic: Dopaminergic; MBEC: minimum biofilm eradication concentration; FO: Fosfomycin; BHI: brain heart infusion; *S. aureus: Staphylococcus aureus*; ppm: parts per million; *P. Subcordiformis*: *Platymonas Subcordiformis*; *P. helgolandica: Platymonas helgolandica*; EC50: half-maximal effective concentration; Cim: Cimetidine; NIH 3T3: NIH/Swiss mouse embryo fibroblast cell line; μM: micromolar; HCT116: colon cancer cell line; IC50: half-maximal inhibitory concentration; UM-SCC-6: human tongue squamous cell carcinoma cell line; ACC-M: adenoid cystic carcinoma cell line; HUVEC: human umbilical vein endothelial cell; mM: millimolar; MCF-7: Michigan Cancer Foundation 7—human breast metastatic adenocarcinoma cell line; HT-29: human colorectal adenocarcinoma cell line with epithelial morphology NSAID: nonsteroidal anti-inflammatory drug; Sm: spontaneous movement; hpf: hours post-fertilization.

## Data Availability

Not applicable.

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
