# Peer review of "Advances in Concentration Gradient Generation Approaches in a Microfluidic Device for Toxicity Analysis"

_cells, 2022, doi:10.3390/cells11193101_

Round 1
Reviewer 1 Report
The paper should be resubmitted after eliminating the part about how articles were selected, etc. all the 2 Section and 3.1 should be removed. If necessary, you could make a supplementary file with it.
Moreover, before resubmission I have several other suggestions:
-Abstract: I would suggest writing the abstract differently. You can move the number of articles and criteria of search in the introduction part but do not emphasize this aspect, but focus on concentration gradient generation approaches in a microfluidic device for toxicity analysis
Please check the grammar. For ex. for lines 461-464" One study [23] concluding that Chrorella is more resistant than Pyraminmonas sp to NaClO, and the others indicating the greater resistance of P. subcordiformis and P. helgolandica var. tsingtoaensis to all metals tested, especially CuSO4, which was shown to be the most toxic"
Author Response
Reviewer #1
- The paper should be resubmitted after eliminating the part about how articles were selected, etc. all the 2 Section and 3.1 should be removed. If necessary, you could make a supplementary file with it.
Answer: Thank you for your time and dedication in reviewing the manuscript. However, our study on systematic review followed the Preferred Reporting Items for Systematic reviews and Meta-Analyses (PRISMA) guideline[1] that helps authors improve the reporting of systematic reviews and meta-analyses, preparing a transparent, complete, and accurate report on why the review was done, what the authors did, and what they found, using advances methods to identify, select, appraise, and synthesize studies. This guideline consists of a 27-item checklist, that details reporting recommendations for each item, and revised flow diagrams for original and updated reviews.
Section 2 (Method) and subsection 3.1 of the Results section were made following all 15 items of the PRISMA checklist of the method section and the 16th item of the result section, as also reported by other studies on a systematic review of MDPI and other journals, highlighting the importance of this detailed description of all the methodological steps with systematic rigor of search, screening, selection process, and data collection, also evaluating the risk of bias of the study and the evaluator in item 2 (method) and introducing the results of the search, and selection process and data organization using a flow diagram, as recommended (3.1 Section).
In a narrative review there are no items that the reviewer recommends being removed, but in a systematic review they are mandatory. We are sending the reference of systematic review articles where it can be corroborated that the items that were described in the present manuscript follow the requirements of a systematic review [2-8].
References
- Liberati A, et al. The PRISMA statement for reporting systematic reviews and meta-analyses of studies that evaluate healthcare interventions: explanation and elaboration. 2009;339:b2700 doi: 10.1136/bmj.b2700 %J BMJ[published Online First: Epub Date].
- Alves AH, et al. The Advances in Glioblastoma On-a-Chip for Therapy Approaches. Cancers 2022;14(4) doi: 10.3390/cancers14040869[published Online First: Epub Date].
- ElAbd R, et. Al, O. Autologous Versus Alloplastic Reconstruction for Patients with Obesity: A Systematic Review and Meta-analysis. Aesthetic Plastic Surgery 2022;46(2):597-609 doi: 10.1007/s00266-021-02664-y[published Online First: Epub Date].
- Rego GNA, et al. Current Clinical Trials Protocols and the Global Effort for Immunization against SARS-CoV-2. Vaccines 2020;8(3) doi: 10.3390/vaccines8030474[published Online First: Epub Date].
- Morrow A, et al. The design, implementation, and effectiveness of intervention strategies aimed at improving genetic referral practices: a systematic review of the literature. Genetics in Medicine 2021;23(12):2239-49 doi: 10.1038/s41436-021-01272-0[published Online First: Epub Date].
- Yao M, et al. The Exploration of Fetal Growth Restriction Based on Metabolomics: A Systematic Review. 2022;12(9):860
- Aguiar M, , et al. mHealth Apps Using Behavior Change Techniques to Self-report Data: Systematic Review. 2022;10(9):e33247 doi: 10.2196/33247[published Online First: Epub Date].
- Flemming N, , et al. Mitochondrial Dysfunction in Individuals with Diabetic Kidney Disease: A Systematic Review. 2022;11(16):2481
Moreover, before resubmission I have several other suggestions:
- Abstract: I would suggest writing the abstract differently. You can move the number of articles and criteria of search in the introduction part but do not emphasize this aspect, but focus on concentration gradient generation approaches in a microfluidic device for toxicity analysis
Answer: The PRISMA guideline has an exclusive section for the structured abstract, indicating the writing in topics, containing objective, data source, selection of studies, data extraction, and synthesis of data, and the conclusion [1]. Considering the methodological importance of the analyses of the manuscript of systematic review, we chose to keep the descriptive part in the abstract, in the same way as the systematic review works, mentioned above, also describe their abstracts in a structured way, following the guideline [2-10].
Reference
- Liberati A, et al. The PRISMA statement for reporting systematic reviews and meta-analyses of studies that evaluate healthcare interventions: explanation and elaboration. 2009;339:b2700 doi: 10.1136/bmj.b2700 %J BMJ.
- Alves AH, et al. The Advances in Glioblastoma On-a-Chip for Therapy Approaches. Cancers 2022;14(4) doi: 10.3390/cancers14040869.
- ElAbd R, et. Al, O. Autologous Versus Alloplastic Reconstruction for Patients with Obesity: A Systematic Review and Meta-analysis. Aesthetic Plastic Surgery 2022;46(2):597-609 doi: 10.1007/s00266-021-02664-y.
- Rego GNA, et al. Current Clinical Trials Protocols and the Global Effort for Immunization against SARS-CoV-2. Vaccines 2020;8(3) doi: 10.3390/vaccines8030474.
- Morrow A, et al. The design, implementation, and effectiveness of intervention strategies aimed at improving genetic referral practices: a systematic review of the literature. Genetics in Medicine 2021;23(12):2239-49 doi: 10.1038/s41436-021-01272-0
- Yao M, et al. The Exploration of Fetal Growth Restriction Based on Metabolomics: A Systematic Review. 2022;12(9):860
- Aguiar M, , et al. mHealth Apps Using Behavior Change Techniques to Self-report Data: Systematic Review. 2022;10(9):e33247 doi: 10.2196/33247
- Flemming N, , et al. Mitochondrial Dysfunction in Individuals with Diabetic Kidney Disease: A Systematic Review. 2022;11(16):2481
- Keeling NJ, et.al. Approaches to assessing the provider experience with clinical pharmacogenomic information: a scoping review. Genetics in Medicine 2021;23(9):1589-603 doi: 10.1038/s41436-021-01186-x.
- Mikat-Stevens NA, et. al. Primary-care providers’ perceived barriers to integration of genetics services: a systematic review of the literature. Genetics in Medicine 2015;17(3):169-76 doi: 10.1038/gim.2014.101
- Please check the grammar. For ex. for lines 461-464" One study [23] concluding that Chrorella is more resistant than Pyraminmonas sp to NaClO, and the others indicating the greater resistance of P. subcordiformis and P. helgolandica var. tsingtoaensis to all metals tested, especially CuSO4, which was shown to be the most toxic"
Answer: Thank you for your observation. We corrected the grammar error.
- English language and style: (x) Extensive editing of English language and style required
Answer: Thank you for your observation. We made all necessary corrections, and an English native speaker reviewed the manuscript.
Reviewer 2 Report
This paper is a well done review on microfluidic devices that use CGG for toxicity assessment in medical and environmental applications. There are still minor revisions that must be done before the paper be in conditions for publication.
1 - Table 3 and 4 are not refereed in the manuscript text. Correct this.
2 - In the abstract line 18 and 19 , is it concentration gradient generator or concentration generator gradient ? Please use just one.
3 - reference 6 is incomplete.
4 - page 4 line 190, here you should tell that this analysis is written in tables. Otherwise, when we read "each table", the reader could be confused.
5 - page 6 line 209 , introduce the table number.
6 - page 7 line 264, in this sentence it is missing some word. Please correct.
7 - page 7, line 269, here when you refer the shape of the microchannels, you should not only write the references, but if there is a figure in figure 2 that reports this situation it should be written. For example, in line 272, write "serpentine channels, figure 2D [41]". And the same for the other situation that a figure exists in figure 2.
8 - page 8, line 325 and 331 and page 11 line 422, also here if there is a figure in figure 2 that reports this situation it should be written.
9 - page 34, line 594, here you should include also a more recent review paper about 3d printing techniques and their applications, for example: https://doi.org/10.3390/s21093304
10- page 34, line 597, there is a paper https://doi.org/10.3390/mi5030738
that shows that photolithography with SU-8 molds can be low-cost and cleanroom less. Please write this option for the readers know and refer that paper.
11 - page 37, line 749, here you should introduce a very recent review paper about the integration of spheroids and organoids into organ-on-a-chip platforms
Author Response
Reviewer #2
This paper is a well done review on microfluidic devices that use CGG for toxicity assessment in medical and environmental applications. There are still minor revisions that must be done before the paper be in conditions for publication.
- Table 3 and 4 are not refereed in the manuscript text. Correct this.
Answer: Thank you for your observation. We added the information about this and citations from Tables 3 and 4 in the respective items 3.4 and 3.5 of the results sections of the manuscript.
- In the abstract line 18 and 19, is it concentration gradient generator or concentration generator gradient ? Please use just one.
Answer: Thank you for your observation. The correct is “Concentration Gradient Generator”, we corrected this in the abstract and reviewed this information over the manuscript.
- reference 6 is incomplete.
Answer: Thank you for your observation. This error is already resolved on this new version of the manuscript.
- page 4 line 190, here you should tell that this analysis is written in tables. Otherwise, when we read "each table", the reader could be confused.
Answer: Thank you for your observation and suggestion. We corrected the phrase (page 4 line 190) to turn clearer.
- page 6 line 209 , introduce the table number.
Answer: Thank you for your suggestion. We changed the phrase (page 6 line 209), introducing the table number.
- page 7 line 264, in this sentence it is missing some word. Please correct.
Answer: Thank you for your observation. We corrected the sentence (page7 line 264) of the manuscript.
- page 7, line 269, here when you refer the shape of the microchannels, you should not only write the references, but if there is a figure in figure 2 that reports this situation it should be written. For example, in line 272, write "serpentine channels, figure 2D [41]". And the same for the other situation that a figure exists in figure 2.
Answer: Thank you for your suggestion. We added in this paragraph (page 7 line 269) of the manuscript the items of Figure 2 that represented the CGG system type cited.
- page 8, line 325 and 331 and page 11 line 422, also here if there is a figure in figure 2 that reports this situation it should be written.
Answer: Thank you for your suggestion. We analyzed the information in these parts (page 8 lines from 325 to 331 and page 11 line 422) of the manuscript and added the items of Figure 2 that represented the CGG system type cited.
- page 34, line 594, here you should include also a more recent review paper about 3d printing techniques and their applications, for example: https://doi.org/10.3390/s21093304
Answer: Thank you for your suggestion. We added more characteristics and information about 3D printing, highlighted in this recent review, and include this reference suggested.
- page 34, line 597, there is a paper https://doi.org/10.3390/mi5030738
that shows that photolithography with SU-8 molds can be low-cost and cleanroom less. Please write this option for the readers know and refer that paper.
Answer: Thank you for your suggestion. We added this interesting information about the optimization of SU-8 molds fabrication in the low-cost process mentioned in the paper [1].
References
- Pinto, V.C.; et al.. Optimized SU-8 Processing for Low-Cost Microstructures Fabrication without Cleanroom Facilities. Micromachines 2014, 5, 738-755.
- page 37, line 749, here you should introduce a very recent review paper about the integration of spheroids and organoids into organ-on-a-chip platforms
Answer: Thank you for your suggestion. We added a recent review about the integration of spheroids and organoids into organ-on-a-chip platforms according to your suggestion.
Reviewer 3 Report
A very interesting paper addressing Advances in concentration gradient generation approaches in a microfluidic device for toxicity analysis.
The paper in my opinion is a good piece of work with conclusions supported by a PRISMA-systematic review (SR) and satisfactory bibliography.
I have some questions and suggestions for paper improvement:
1- Please verify your work for English and mistyping errors.
2- Please correct inconsistencies in nomenclature and double check references
3- A Systematic Review aims to collect all empirical evidence that fits pre-specified eligibility criteria to answer a precise research query. SRs are regularly used in clinical research and social sciences; however, they have found application in various other subject areas ex. environmental sciences, engineering and basic science research. Thus, it is correct to use Prisma-SR in this area of microfluidics.
My question is: Do you think the Scopus & PubMed data bases enough to obtain sufficient information to deliver a clear and comprehensive overview of available evidence on the present subject?
4- Having obtained the SR outcomes authors performed extraction of information in the following areas: - Characteristics, design and fabrication of Concentration Gradient Generators microfluidic devices for toxicity analyses, - Concentration Gradient Generators characteristics of microfluidic device, Biological model used to toxicity evaluation in the CGG microfluidic device & Toxicity screening evaluation, and outcome of the CGG microfluidic device; all these information was formated in four tables.
My question is: It is possible to identify research gaps and methodological problems in the current understanding of a field
5-SR is a very interesting technique, on the other hand there is a risk of not considering very new materials or techniques, see: Monjezi, M.; Rismanian, M.; Jamaati, H.; Kashaninejad, N. Anti-Cancer Drug Screening with Microfluidic Technology. Appl. Sci. 2021, 11, 9418. https://doi.org/10.3390/app11209418
Please comment
6- PDMS is not a perfect material It has four main drawbacks (DOI: 10.21037/mps.2018.08.02): A) Absorption. The porous nature of PDMS enables small hydrophobic molecules to diffuse into the bulk polymer, it also absorbs hydrocarbon solvents and swells up like a sponge. B) Leaching. This issue caused by uncrosslinked oligomers. C) Unstable wettability. In general, PDMS devices would be rendered hydrophilic by using oxygen plasma for further operation. D) Poor chemical compatibility. The dissolution of PDMS in organic solvent would cause swelling. Due to the pointed drawbacks several researchers are recently working with other materials as perfluorinated polymers , COC and others as better substrates for CGG devices.
My question is: Why PRISMA-SR fail to point out the new directions in materials for CGG fabrication?
7- The CGG system is a faster, cheap and more accurate method for drug and chemical pollutants toxicity analysis. You have pointed out from the SR results generators of a concentration gradient, by convection mixing-based, laminar flow diffusion-based (Y-shape), membrane-based, pressure balance-based, droplet-based, and flow-based methods.
The droplet-based methods That seems to be one of the future research directions are mainly divided into three types through droplet generation, droplet coalescence, and droplet dilution respectively.
Surprisingly SR provides only one article on the subject. What is the cause of this misleading information?
8- SR has the advantage of identify questions for which the available evidence provides clear answers so no further research is necessary.
Have you identified some points where research is no more necessary with the information you retrieved from your PRISMA-SR?
Author Response
Reviewer #3
A very interesting paper addressing Advances in concentration gradient generation approaches in a microfluidic device for toxicity analysis.
The paper in my opinion is a good piece of work with conclusions supported by a PRISMA-systematic review (SR) and satisfactory bibliography.
I have some questions and suggestions for paper improvement:
1- Please verify your work for English and mistyping errors.
Answer: Thank you for your observation. We made all necessary corrections, and an English native speaker reviewed the manuscript.
2- Please correct inconsistencies in nomenclature and double check references
Answer: Thank you for your observation. We corrected all inconsistencies related to nomenclature and references.
3- A Systematic Review aims to collect all empirical evidence that fits pre-specified eligibility criteria to answer a precise research query. SRs are regularly used in clinical research and social sciences; however, they have found application in various other subject areas ex. environmental sciences, engineering and basic science research. Thus, it is correct to use Prisma-SR in this area of microfluidics.
My question is: Do you think the Scopus & PubMed data bases enough to obtain sufficient information to deliver a clear and comprehensive overview of available evidence on the present subject?
Answer: Both databases are worldwide known and used in systematic review, have characteristics complementary that provide a high capacity to identify the main studies. The database PubMed is incredibly practical, rapid, and simple to use. It is the most often utilized source of information in the biomedical sector and health fields, and related disciplines such as life sciences, behavioral sciences, chemical sciences, and bioengineering. Due to its availability for free and provide open access to all interested clinicians, researchers, and trainees and to the public in general, besides containing more than 34 million citations and abstracts of biomedical literature. In comparison to PubMed and Web of Science, Scopus already offers a wider range of journals, and its citation analysis is faster and covers a larger number of articles. PubMed originate from the United States, whereas Scopus originates from Europe. PubMed focuses mainly on medicine and biomedical sciences, whereas Scopus cover most scientific fields. PubMed allows the larger number of keywords per search but is the only database that does not provide citation analysis, available to the public online since 1996, was developed and is maintained by the National Center for Biotechnology Information (NCBI), at the U.S. National Library of Medicine (NLM), located at the National Institutes of Health (NIH). Scopus includes articles published from 1966 on, but information regarding citation analysis is available only for articles published after 1996. One major advantage of PubMed, not reproduced by Scopus is that it is readily updated not only with printed literature but also with literature that has been presented online in an early version before print publication by various journals. In contrast, Scopus is readily updated for printed literature but do not include online early versions. PubMed was developed by the National Library of Medicine, a division of the National Institutes of Health, and rapidly became syn-onymous with medical literature research worldwide. The Scopus database was developed by Elsevier, combining the characteristics of both PubMed and Web of Science. These combined characteristics allow for enhanced utility, both for medical literature research and academic needs (citation analysis).
4- Having obtained the SR outcomes authors performed extraction of information in the following areas: - Characteristics, design and fabrication of Concentration Gradient Generators microfluidic devices for toxicity analyses, - Concentration Gradient Generators characteristics of microfluidic device, Biological model used to toxicity evaluation in the CGG microfluidic device & Toxicity screening evaluation, and outcome of the CGG microfluidic device; all these information was formated in four tables.
My question is: It is possible to identify research gaps and methodological problems in the current understanding of a field
Answer: Yes, the selected articles no reported possible interferences of the interaction between in the material used in the device fabrication with biological model or substance tested in the toxicity screening, considering the advantages and disadvantages on variety materials used [1,2]. Another important methodological aspect that few studies adopted was the simulation procedure before device fabrication that contribute to determine the best geometric, physics and biologic parameters for the device development for toxicity screening using CGG technology. Methodological careful regarding the stability duration of concentrations obtained of the drugs into CGG system did not report in majority of studies, gap that may compromise the accuracy of toxicity evaluation. These information’s also were added in the discussion section of the manuscript.
5- SR is a very interesting technique, on the other hand there is a risk of not considering very new materials or techniques, see: Monjezi, M.; Rismanian, M.; Jamaati, H.; Kashaninejad, N. Anti-Cancer Drug Screening with Microfluidic Technology. Appl. Sci. 2021, 11, 9418. https://doi.org/10.3390/app11209418
Answer: Thank you for your observation. We agree that the disadvantages of PDMS and new materials that are gaining relevance for the fabrication of microfluidic devices should have been explored. We included sections on both topics in the Discussion.
Please comment
6- PDMS is not a perfect material It has four main drawbacks (DOI: 10.21037/mps.2018.08.02): A) Absorption. The porous nature of PDMS enables small hydrophobic molecules to diffuse into the bulk polymer, it also absorbs hydrocarbon solvents and swells up like a sponge. B) Leaching. This issue caused by uncrosslinked oligomers. C) Unstable wettability. In general, PDMS devices would be rendered hydrophilic by using oxygen plasma for further operation. D) Poor chemical compatibility. The dissolution of PDMS in organic solvent would cause swelling. Due to the pointed drawbacks several researchers are recently working with other materials as perfluorinated polymers , COC and others as better substrates for CGG devices.
My question is: Why PRISMA-SR fail to point out the new directions in materials for CGG fabrication?
Answer: The PRISMA-SR normally highlight the relevant aspects of issue, given to reader the overview the aspects most used, but not necessary the most adequate, failing in the corrected direction of the best material of device fabrication, since the PDMS even so the most used, has a lot of disadvantages in the CGG fabrication for drug toxicity screening, mainly in the translational research and commercial production in large scale.
7- The CGG system is a faster, cheap and more accurate method for drug and chemical pollutants toxicity analysis. You have pointed out from the SR results generators of a concentration gradient, by convection mixing-based, laminar flow diffusion-based (Y-shape), membrane-based, pressure balance-based, droplet-based, and flow-based methods.
The droplet-based methods That seems to be one of the future research directions are mainly divided into three types through droplet generation, droplet coalescence, and droplet dilution respectively.
Surprisingly SR provides only one article on the subject. What is the cause of this misleading information?
Answer: Undoubtedly, the generation by diffusion using the droplet method has a good future perspective regarding the advances of microfluidics and the generation of concentration gradient [3]. However, there is a report on the difficulty of controlling flow and concentration while maintaining the droplet shape. [4-6]. Such parameters, however, are crucial for the assessment of toxicity (target point of our SR), thus resulting in a single reported study [7].
8- SR has the advantage of identify questions for which the available evidence provides clear answers so no further research is necessary.
Have you identified some points where research is no more necessary with the information you retrieved from your PRISMA-SR?
Answer: This systematic review was able to identify the homogeneity of the findings in relation to the manufacture of microfluidic devices in the last 10 years, while the technique, materials used and sealing methods, allowing the identification of other techniques, materials and methods that can also be used aiming the same goal [8,9]. Furthermore, the methods used to generate concentration gradients, convection and fluid diffusion are well described in the literature. In the same way as it became clear that the method used is intrinsically associated with the structure used, where a single device, due to its structure, can present the two associated methods in a single device due to its design. [3,10,11]. And all these characteristics will depend on the objective proposed by the author of each study individually, and may be associated with new technologies such as valves [12], and vacuum systems [13] also identified in this work.
References
- Ren, K.; Zhou, J.; Wu, H. Materials for Microfluidic Chip Fabrication. Accounts of Chemical Research 2013, 46, 2396-2406, doi:10.1021/ar300314s.
- Nielsen, J.B.; Hanson, R.L.; Almughamsi, H.M.; Pang, C.; Fish, T.R.; Woolley, A.T. Microfluidics: Innovations in Materials and Their Fabrication and Functionalization. Anal Chem 2020, 92, 150-168, doi:10.1021/acs.analchem.9b04986.
- Wang, X.; Liu, Z.; Pang, Y. Concentration gradient generation methods based on microfluidic systems. RSC Advances 2017, 7, 29966-29984, doi:10.1039/C7RA04494A.
- Cao, J.; Kürsten, D.; Schneider, S.; Knauer, A.; Günther, P.M.; Köhler, J.M. Uncovering toxicological complexity by multi-dimensional screenings in microsegmented flow: modulation of antibiotic interference by nanoparticles. Lab on a Chip 2012, 12, 474-484, doi:10.1039/C1LC20584F.
- Kaminski, T.S.; Jakiela, S.; Czekalska, M.A.; Postek, W.; Garstecki, P. Automated generation of libraries of nL droplets. Lab on a Chip 2012, 12, 3995-4002, doi:10.1039/C2LC40540G.
- Sjostrom, S.L.; Joensson, H.N.; Svahn, H.A. Multiplex analysis of enzyme kinetics and inhibition by droplet microfluidics using picoinjectors. Lab on a Chip 2013, 13, 1754-1761, doi:10.1039/C3LC41398E.
- Zeng, W.; Chen, P.; Li, S.; Sha, Q.; Li, P.; Zeng, X.; Feng, X.; Du, W.; Liu, B.-F. Hand-powered vacuum-driven microfluidic gradient generator for high-throughput antimicrobial susceptibility testing. Biosensors and Bioelectronics 2022, 205, 114100, doi:https://doi.org/10.1016/j.bios.2022.114100.
- Scott, S.M.; Ali, Z. Fabrication Methods for Microfluidic Devices: An Overview. Micromachines (Basel) 2021, 12, doi:10.3390/mi12030319.
- Tiwari, S.K.; Bhat, S.; Mahato, K.K. Design and Fabrication of Low-cost Microfluidic Channel for Biomedical Application. Scientific Reports 2020, 10, 9215, doi:10.1038/s41598-020-65995-x.
- Williams, I.; Lee, S.; Apriceno, A.; Sear, R.P.; Battaglia, G. Diffusioosmotic and convective flows induced by a nonelectrolyte concentration gradient. Proc Natl Acad Sci U S A 2020, 117, 25263-25271, doi:10.1073/pnas.2009072117.
- Kim, H.J. Biomimetic Microengineering; CRC Press: 2020.
- Zheng, G.; Wang, Y.; Wang, Z.; Zhong, W.; Wang, H.; Li, Y. An integrated microfluidic device in marine microalgae culture for toxicity screening application. Mar Pollut Bull 2013, 72, 231-243, doi:10.1016/j.marpolbul.2013.03.035.
- Zeng, W.; Chen, P.; Li, S.; Sha, Q.; Li, P.; Zeng, X.; Feng, X.; Du, W.; Liu, B.F. Hand-powered vacuum-driven microfluidic gradient generator for high-throughput antimicrobial susceptibility testing. Biosens Bioelectron 2022, 205, 114100, doi:10.1016/j.bios.2022.114100.
Round 2
Reviewer 1 Report
Given the fact that this work is accordingly to the Systematic reviews and Meta-Analyses (PRISMA) guidelines, and this is accepted by the journal, I have no further comments, therefore the paper can be accepted .
Reviewer 3 Report
Dear authors, after reading the new paper version, I agree with your modifications. Therefore, I recommend this version for publication.